# Study of Training Dynamics for Memory-Constrained Fine-Tuning

**Aël Quélennec, Nour Hezbri, Pavlo Mozharovskyi, Van-Tam Nguyen & Enzo Tartaglione**
LTCI, Télécom Paris
Palaiseau, France
`{name}.{surname}@telecom-paris.fr`

## Abstract

Memory-efficient training of deep neural networks has become increasingly important as models grow larger while deployment environments impose strict resource constraints. We propose TraDy, a novel transfer learning scheme leveraging two key insights: layer importance for updates is architecture-dependent and determinable a priori, while dynamic stochastic channel selection provides superior gradient approximation compared to static approaches. We introduce a dynamic channel selection approach that stochastically resamples channels between epochs within preselected layers. Extensive experiments demonstrate TraDy achieves state-of-the-art performance across various downstream tasks and architectures while maintaining strict memory constraints, achieving up to 99% activation sparsity, 95% weight derivative sparsity, and 97% reduction in FLOPs for weight derivative computation.

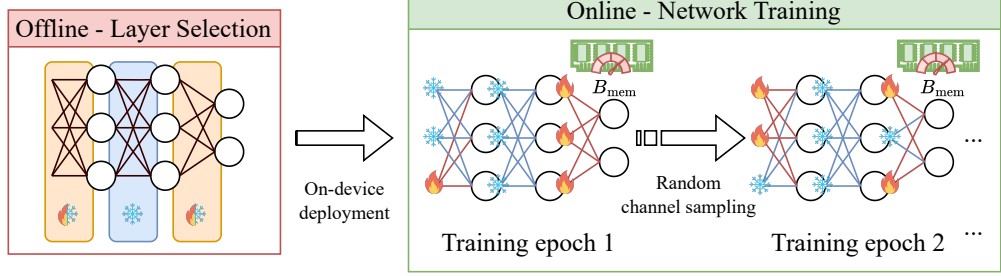

Figure 1: TRady dynamically reselects the subgraph to update within the memory budget $B_{\mathrm{mem}}$.

## 1 Introduction

In the span of a decade, machine and deep learning have become key technologies in the computer science landscape. They have found a wide variety of practical applications in fields such as Natural Language Processing Vaswani et al. (2017); Tenney et al. (2019), Computer Vision Krizhevsky et al. (2012); Simonyan & Zisserman (2014); Minaee et al. (2021), or Speech Recognition Deng et al. (2013); Nassif et al. (2019). This surge in popularity can be largely explained by the ever-increasing performances of new architectures, intimately linked to hardware innovations Baji (2018). The design of better-performing parallel computing units (such as GPUs and TPUs) allows the training of large neural networks that feature increasing overparameterization compared to their predecessors Sevilla et al. (2022). If this trend further demonstrates deep learning principles' innate generalization potential, it raises ecological and technical concerns. Training and exploitation of these architectures require very high energy consumption, and their deployment in real-world environments is impossible without extensive compression, leading to performance worsening.

The research field of efficient neural network compression, consequently, has gained a surge of interest in recent years. The main pillars of this research area are quantization, low-rank compression, efficient design of compact models, knowledge distillation, and network sparsification (also known as pruning) Cheng et al. (2018); Deng et al. (2020). These methods aim to optimize the trade-off between memory/energy consumption and the inference accuracy of models in resource-constrained

environments. However, inference is only part of the life cycle of a deep neural network, and these works do not provide solutions to perform memory-efficient training. As such, compressed models trained offline and deployed on-device suffer from a phenomenon called *data drift* Sahiner et al. (2023) which results in performance degradation over time. Alternatively, enabling on-device learning would improve the viability and efficiency of embedded AI through several use cases, including user adaptivity or lifelong learning Incel & Bursa (2023).

One major obstacle to making on-device learning practical is the computational and memory burden of backpropagation. For embedded devices, limited memory and computational capacity create hard constraints that cannot be exceeded. Some methods attempt to address memory limitations by exploring alternatives to backpropagation, like hyperdimensional computing for tasks like image segmentation Yang et al. (2023a), the Forward-Forward algorithm Hinton (2022), and PEPITA Pau & Aymone (2023). Although these strategies are promising, they generally don't match the performance of backpropagation-based techniques. One direct approach attempting to solve memory and computation issues was proposed by Lin et al. (2022), where a static subnetwork is updated for any downstream task. Orthogonally, Yang et al. (2023b) and more recently Nguyen et al. (2024) reduce memory consumption by compressing the elements to store for backpropagation. Each approach, however, comes with its own limitations—either compromising accuracy or introducing additional latency.

We propose **Tra**ining **Dy**namics (TraDy) for memory-constrained transfer learning. Given a pre-trained network, we make three key propositions regarding the training dynamics when performing memory-constrained transfer learning on a downstream task. Building on top of these, we design a dynamic channel selection algorithm for efficient transfer learning under memory constraints (Fig. 1). Our main contributions can be summarized as follows.

- We show that stochastic gradients exhibit heavy-tailed behavior during transfer learning, creating natural sparsity patterns that facilitate efficient gradient pruning (Sec. 3.2).

- We show that the relative importance of network layers remains consistent across downstream tasks and primarily depends on network architecture rather than task specifics, enabling a priori layer selection (Sec. 3.3).

- We establish that channel importance distributions within layers are task-dependent and cannot be predetermined without task data, while calculating importance metrics for all channels contradicts on-device memory constraints (Prop. 3.2 and Sec. 4.1).

- We introduce TraDy, a dynamic stochastic channel selection approach that resamples channels between epochs within pre-selected layers, effectively approximating the full gradient while maintaining strict memory constraints (Sec. 3.4).

- Our experiments illustrate that TraDy achieves state-of-the-art performance in various downstream tasks and network architectures while respecting memory limitations through high levels of both weight and activation sparsities alongside reduced FLOPs, validating our theoretical insights (Sec. 4.2).

## 2 RELATED WORKS

**Gradient Pruning.** Sub-network selection for training, whether static or dynamic, can be referred to as gradient pruning. Unlike classical pruning, gradient pruning preserves the complete network during inference, only modifying the backpropagation phase by selectively computing gradients based on specific criteria. While gradient pruning in on-device learning primarily addresses memory constraints, other applications focus on accelerating training with minimal accuracy impact Zhang et al. (2024); Bragagnolo et al. (2022); Li et al. (2023); Ye et al. (2020); McDanel et al. (2022). Particularly relevant to our fine-tuning approach is Lee et al. (2022), who explore gradient pruning as a regularization technique. They demonstrate that network blocks can contribute either positively or negatively to downstream task performance, creating task-specific optimal configurations for selective updating. Their work shows that the ratio of gradient norm to parameter norm effectively predicts which blocks should be updated or frozen for optimal transfer learning performance.

**On-Device Learning.** Our work draws inspiration from three key contributions in the on-device learning domain, where memory and energy constraints necessitate efficient fine-tuning of pre-trained models rather than training from scratch.

Lin et al. (2022) introduced Sparse Update (SU) schemes, a selective parameter updating strategy

that enables fine-tuning on extreme edge devices along with operator reordering and quantization-aware scaling. Their approach demonstrated that memory-efficient subnetworks can yield acceptable performance on downstream tasks. However, finding adequate SU schemes requires heavy pre-computation through offline accuracy contribution analysis, followed by evolutionary search for each network and memory budget. Additionally, SU applies uniformly across all downstream tasks, implicitly assuming that selected layers and channels are optimal for each individual task and should remain fixed throughout training.

Building on this foundation, Kwon et al. (2024) improved adaptability to new architectures, datasets, and memory budgets. Their approach ranks layers by computing Fisher information on activations from downstream task samples, then applies reweighting by parameter count and MAC operations. Despite increased flexibility, computing Fisher information for all network channels requires more memory than gradient computation itself, contradicting the original memory constraints. Like SU, this approach still employs static selection that does not adapt during training.

Quélennec et al. (2024) propose dynamic subnetwork selection between epochs using a "velocity" metric that quantifies neuron output changes when fed with consistent data. Their results demonstrated accuracy improvements over static selection within fixed parameter budgets. While promising and flexible across networks and datasets, this method is limited by its exclusive focus on parameter count without considering activation memory, which represents an equally significant constraint in on-device scenarios Cai et al. (2020).

Our work builds upon these foundations by analyzing transfer learning dynamics in deep neural networks and demonstrating how a theoretically-grounded dynamic channel selection strategy can overcome limitations of previous approaches while maintaining strict memory constraints.

## 3 METHOD

In this section, we unfold our study towards parameter-efficient fine-tuning under extreme memory constraints. After formulating our problem in Sec. 3.1, we introduce the theoretical foundations of heavy-tailed gradient distributions and our memory-aware gradient norm metric in Sec. 3.2. This theoretical framework guides our analysis of layer behavior in Sec. 3.3, where we demonstrate the architecture-dependent nature of layer importance. Building on these insights, we introduce our dynamic channel sampling strategy in Sec. 3.4, which enables efficient transfer learning within strict memory budgets by stochastically resampling channels between epochs from pre-selected layers.

### 3.1 PROBLEM FORMULATION AND NOTATIONS

Our goal is to fine-tune a pre-trained neural network on a downstream task under specific memory constraints, without prior knowledge of the target task. Although the target device can execute the complete forward pass, the memory limitations prevent training all network parameters simultaneously. Therefore, we aim to strategically select which portions of the architecture to train, optimizing performance while keeping the combined weight and activation memory within the specified budget. Our analysis focuses specifically on standard 2D convolutions within Convolutional Neural Networks (CNNs), excluding bias terms.[1] The CNN then writes as a sequence of $n$ convolutional layers:

$$\mathcal{F}(\mathcal{X}) = (\mathcal{C}_{\mathcal{W}_n} \circ \mathcal{C}_{\mathcal{W}_{n-1}} \circ \cdots \circ \mathcal{C}_{\mathcal{W}_2} \circ \mathcal{C}_{\mathcal{W}_1})(\mathcal{X}), \tag{1}$$

with $\mathcal{X}$ the input of the network and $\mathcal{W}_i \in \mathbb{R}^{C' \times C \times D \times D}$ the weight kernels, $C$ and $C'$ the number of input and outputs channels and $D$ the kernel dimensions.

Given the $i$-th layer, we note $\mathcal{A}_i \in \mathbb{R}^{B \times C \times H \times W}$ and $\mathcal{A}_{i+1} \in \mathbb{R}^{B \times C' \times H' \times W'}$ as its input and output activation tensors, where $B$ is the batch size, $H$ and $W$ are the width and height of the feature map. To compute the weight derivatives $\frac{\partial \mathcal{L}}{\partial \mathcal{W}_i}$, the loss $\mathcal{L}$ is calculated at the output of the network and backpropagated to the $i^{th}$ layer through the activation derivatives as $\frac{\partial \mathcal{L}}{\partial \mathcal{A}_{i+1}}$. We then get the weight derivatives:

$$\left[\frac{\partial \mathcal{L}}{\partial \mathcal{W}_i}\right]_{c',c,k,l} = \sum_{b=1}^{B} \sum_{h'=1}^{H'} \sum_{w'=1}^{W'} [\mathcal{A}_i^p]_{b,c,h,w} \left[\frac{\partial \mathcal{L}}{\partial \mathcal{A}_{i+1}}\right]_{b,c',h',w'}, \tag{2}$$

where $h = h' \times \text{stride} + k \times \text{dilation}$, $w = w' \times \text{stride} + l \times \text{dilation}$ and $\mathcal{A}_i^p$ is the padded input. Similar to unstructured pruning, removing individual parameters does not yield significant compu-

---

[1] A similar analysis can be conducted for fully-connected layers.

tational and memory gains, as it creates inefficient unstructured sparse tensors. A more effective approach involves freezing along specific weight dimensions, enabling efficient tensor operations and creating structured gradient sparsity Bragagnolo et al. (2022). When considering selective freezing, we have four potential dimensions: input channels, output channels, and the two kernel dimensions. However, these options differ significantly in their effectiveness for memory optimization. While freezing along output channels reduces memory needed for storing activation derivative tensors, these derivatives must still be fully computed to ensure accurate gradient propagation through subsequent layers. After analyzing all possibilities, freezing along the input channels dimension emerges as the only approach that simultaneously achieves both **weight sparsity** and **activation sparsity**.

From Eq. 2, we observe that updating an input channel $c$ requires storing only the corresponding activation values in memory. Specifically, when freezing weight tensors along the input channel dimension, the gradient components form natural groupings that can be treated as independent units. This dual benefit eliminates both the storage requirements for the corresponding activations and the computational burden of calculating their associated weight gradients, making input channel freezing optimal for memory-constrained scenarios.

Based on Eq. 2, we derive analytical expressions for both memory requirements and computational complexity associated with updating a single input channel $c$ within layer $i$ for a single data input. Let $\mathcal{C}_c^{\mathcal{W}_i} = C' \times D \times D$ represent the weight memory cost and $\mathcal{C}_c^{\mathcal{A}_i} = H \times W$ represent the activation memory cost for channel $c$. The total space complexity $(\Theta_{\text{space}})_c$ and time complexity $(\Theta_{\text{time}})_c$ are:

$$(\Theta_{\text{space}})_c = \mathcal{C}_c^{\mathcal{W}_i} + \mathcal{C}_c^{\mathcal{A}_i}, \tag{3}$$

$$(\Theta_{\text{time}})_c = D^2 C' H' W'. \tag{4}$$

These expressions demonstrate that input channel-level selection provides fine-grained control over both memory and computational consumption while maintaining the structural coherence necessary for effectively exploiting the heavy-tailed sparsity patterns described in the following section, thus achieving the critical combination of weight and activation sparsity essential for memory-efficient fine-tuning.

## 3.2 Heavy-tailed Theory and Gradient Norm Metric

The stochastic nature of gradient descent has significant theoretical implications for our approach. Simsekli et al. (2019) established that stochastic gradient noise follows a heavy-tailed distribution during training with SGD. Such distributions are characterized by a tail-index parameter $\alpha \in (0, 2]$ and exhibit power-law decay proportional to $1/|x|^{\alpha+1}$. When $\alpha = 2$, this distribution reduces to a Gaussian; for all other values of $\alpha$, the resulting random variable has infinite variance. This heavy-tailed noise can be mathematically formulated as:

$$U_k(\mathcal{W}) = \Delta\tilde{\mathcal{W}}_k - \Delta\mathcal{W}, \tag{5}$$

where $\Delta\mathcal{W}$ denotes the true gradient computed using the entire dataset, $\Delta\tilde{\mathcal{W}}_k$ represents the stochastic gradient estimated from $k$ randomly sampled data points, and $U_k$ follows a symmetric $\alpha$-stable distribution $U_k \sim \mathcal{S}\alpha\mathcal{S}(\sigma)$. In this notation, $\sigma$ serves as a scale parameter controlling the distribution's spread around zero.

Building on this foundation, Wan et al. (2023) demonstrated that injecting heavy-tailed noise during weight updates inherently enhances network compressibility for pruning operations. Their key insight reveals that heavy-tailed noise causes the weight matrix columns to follow multivariate heavy-tailed distributions independently of each other. Consequently, the norm distribution becomes highly skewed as a small subset of columns exhibits disproportionately large norms while most remain relatively small. This concentration means that the overall weight matrix norm is mostly determined by just a few dominant columns, creating an implicit structure that aligns perfectly with sparse update requirements.

In our approach, we extend this theoretical framework to the domain of gradient pruning rather than weight pruning. From (5), we can observe that gradients naturally decomposes as the sum of the stochastic gradients and a heavy-tailed noise term $U_k$. Applying the insights from Wan *et al.*, we expect that gradient norms will concentrate disproportionately in a small subset of channels. This creates a natural opportunity for selective gradient computation and parameter updating. To systematically exploit this property, we define the input channel gradient norm as:

$$\left\| \left( \frac{\partial \mathcal{L}}{\partial \mathcal{W}_i} \right)_c \right\|_2 = \sqrt{ \sum_{c',k,l} \left[ \frac{\partial \mathcal{L}}{\partial \mathcal{W}_i} \right]_{c',c,k,l}^2 }. \tag{6}$$

While the raw gradient norm provides valuable information about update importance, it fails to account for the memory constraints that are central to our scenario. To address this limitation, we introduce a memory-aware metric called the Reweighted Gradient Norm (RGN). This metric incorporates both computational significance and memory efficiency by dividing the raw gradient norm by the total memory cost associated with updating that channel. Using the notation established in Sec. 3.1, we define RGN as:

$$\text{RGN}_c = \frac{\left\|\left(\frac{\partial \mathcal{L}}{\partial \mathcal{W}_i}\right)_c\right\|_2}{\mathcal{C}_c^{\mathcal{W}_i} + \mathcal{C}_c^{\mathcal{A}_i}}. \tag{7}$$

This reweighting counteracts the bias toward channels with higher parameter counts as they naturally show larger gradient norms. By directly incorporating memory costs, RGN creates different layers and channels' order compared to the raw gradient norm. It is thus well-suited for memory-constrained settings as it optimizes update efficiency through prioritization of less memory-intensive channels when raw gradient norms are similar. This allows more parameters to be updated within the same memory budget, potentially improving performance per memory unit.

We use this RGN metric throughout our analysis to examine layer and channel importance across different architectures, datasets, and seeds, informing our final solution design.

### 3.3 Layers Behavior During Fine-tuning

Just as heavy-tailed gradient properties create natural sparsity patterns among channels, we hypothesize that similar dynamics may govern layer-level importance. This section explores how gradient norm distribution across layers influences their relative contribution to the fine-tuning process and how this knowledge can guide our parameter selection strategy. We decide to characterize the layer reweighted gradient norm as follows:

$$\left\|\frac{\partial \mathcal{L}}{\partial \mathcal{W}_i}\right\|_{\text{RGN}} = \sum_{c=1}^{C} \text{RGN}_c = \sum_{c=1}^{C} \frac{\left\|\left(\frac{\partial \mathcal{L}}{\partial \mathcal{W}_i}\right)_c\right\|_2}{\mathcal{C}_c^{\mathcal{W}_i} + \mathcal{C}_c^{\mathcal{A}_i}} = \frac{1}{(\Theta_{\text{space}})_i} \sum_{c=1}^{C} \left\|\left(\frac{\partial \mathcal{L}}{\partial \mathcal{W}_i}\right)_c\right\|_2. \tag{8}$$

**Proposition 3.1.** *The relative ranking of layers to their reweighted gradient norm remains largely invariant over time during training and across different downstream tasks. This ranking is primarily determined by the network architecture rather than dataset-specific characteristics.*

Based on neural network architecture, certain layers consistently exhibit higher gradient norms than others. This architectural dependency is particularly evident in networks with residual connections. Skip connections mitigate gradient vanishing by effectively reducing the virtual depth of the network for certain computational paths. As a result, we typically observe a characteristic pattern in the distribution of gradient norms: the first layer of each residual block generally displays a significantly higher gradient norm than subsequent layers within the same block.

We provide a detailed analysis of this phenomenon in the appendix, Sec. C and empirically validate such behavior in Sec. 4.1.

Based on these observations, we can strategically restrict parameter updates to the subset of layers that naturally receive higher gradients. Recent literature supports this approach, with multiple studies demonstrating that selectively updating certain layers provides significant contributions to model optimization on downstream tasks Kaplun et al. (2023); Zhang & Bottou (2024); Lee et al. (2022). The practical implication is substantial: depending on the similarity between pre-training and downstream tasks, updating only a carefully selected subset of layers can maintain performance comparable to full fine-tuning while significantly reducing memory requirements.

### 3.4 Dynamic Channel Sampling

After analyzing layer-level behavior, we now focus on individual input channels within selected layers.

**Proposition 3.2.** *The distribution of channel gradient norms varies between datasets.*

From the weight derivative in (2), two key components emerge: activation maps reflecting network feature extraction and activation derivatives shaping the task-specific loss landscape. Both are fundamentally task-dependent, justifying that channel gradient norms vary between downstream

---

**Algorithm 1** TraDy

---

**Input:** Pre-trained backbone weights $\mathcal{W}$, number of epochs $n$, train data $D_{\text{train}}$, test data $D_{\text{test}}$, memory budget $B_{\text{mem}}$, set of relevant layers $\{L_K\}$.
**Function:**
    **for** epochs $= 1$ **to** $n$ **do**
        Randomly sample channels $\{C^t\}$ within the set of relevant layers $\{L_K\}$ along uniform probability distribution until the memory budget $B_{\text{mem}}$ is met.
        Update weights of the selected channels using $D_{\text{train}}$.
    **end for**
    Evaluate the fine-tuned backbone using $D_{\text{test}}$.

---

tasks. We provide empirical validation in Sec. 4.1.

While Sec. 3.3 establishes that layers can be predetermined architecturally, static channel selection proves inadequate since RGN distributions are task-dependent. In real-world scenarios where downstream data is unavailable offline and memory constraints prevent full gradient computation, directly estimating channel RGN distributions introduces overhead contradicting our efficiency goals. Instead, we propose TraDy, a dynamic sampling strategy operating within memory constraints. Our approach randomly selects input channels to update from the predetermined layers, resampling between epochs while maintaining strict memory budget compliance throughout training. This strategy ensures that the combined activation and weight memory consumption remains strictly below the specified memory budget throughout the training process, effectively balancing exploration of the channel space with the practical constraints of edge devices.

Leveraging layer selection from Sec. 3.3, most gradient information concentrates in selected layers, as predicted by heavy-tailed theory (Sec. 3.2). Random channel selection changing dynamically ensures the expectation of selected gradients approximates the full gradient within efficient layers over time. Let $\Delta\tilde{\mathcal{W}}_t$ denote the non-null gradient at epoch $t$ and $\Delta\mathcal{W}_{\{C^t\}}$ the sparse gradient from randomly selected set of channels $\{C^t\}$ at epoch $t$ within pre-selected et of layers $\{L_K\}$. Following the principle that stochastic gradient expectation equals full gradient expectation, and due to our layer selection excluding low-magnitude gradients while the stochastic channel selection follows a uniform distribution, we have:

$$\mathbb{E}\left[\sum_t \Delta\tilde{\mathcal{W}}\right] \simeq \mathbb{E}\left[\sum_t \Delta\mathcal{W}_{\{C^t\}}\right]. \tag{9}$$

The computational complexity of randomly and successively selecting $k$ elements from $n$ channels is $\mathcal{O}(k\log(n))$, negligible compared to gradient computation itself.

We present here TraDy, our dynamic subnetwork update pipeline for transfer learning, under memory constraints, depicted in Alg. 1. Given a pre-trained backbone and a training dataset, channels are randomly sampled within the fixed set of layers of interest $\{L_K\}$ and updated conditioned on the memory budget (line 4). At the end of the training, we evaluate our model's performance on the test dataset (line 7). In the next section, we will present our empirical results.

## 4 EXPERIMENTS

This section describes the experiment conducted to validate the hypothesis proposed in Sec. 3 as well as compare its performance to other sparse update strategies. A complete description of the experimental setup is proposed in Sec. D.1 of the appendix.

### 4.1 GRADIENT STUDY

**Heavy-Tailed Stochastic Gradient.** We empirically validate the heavy-tailed characteristic of stochastic gradients during fine-tuning, as introduced in Sec. 3.2. Following methodology similar to Şimşekli *et al.*, we use the Mohammadi et al. (2015) estimator for $\alpha$-stable distributions. For each fine-tuning epoch $t$, we collect stochastic gradients of all $P$ trainable parameters across $S$ training steps, constructing a $P \times S$ matrix. This matrix is processed by the estimator to produce $\alpha_t$, representing the heavy-tailed index of the stochastic gradient distribution during epoch $t$. Fig. 2 illustrates the evolution of $\alpha$ for our three network architectures when fine-tuned on three diverse downstream

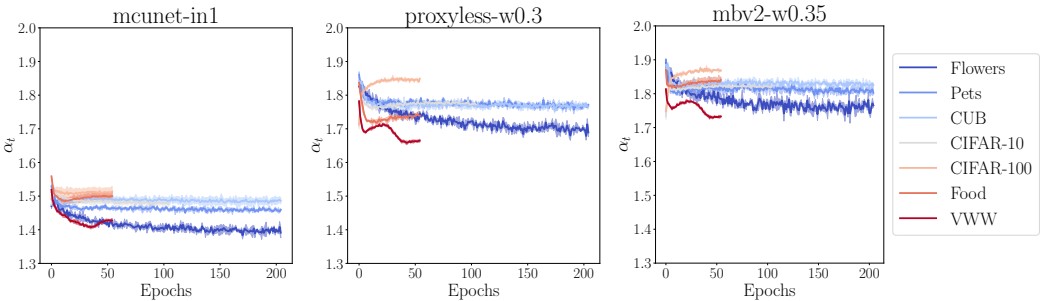

Figure 2: Evolution of stochastic gradient heavy-tailed index $\alpha_t$.

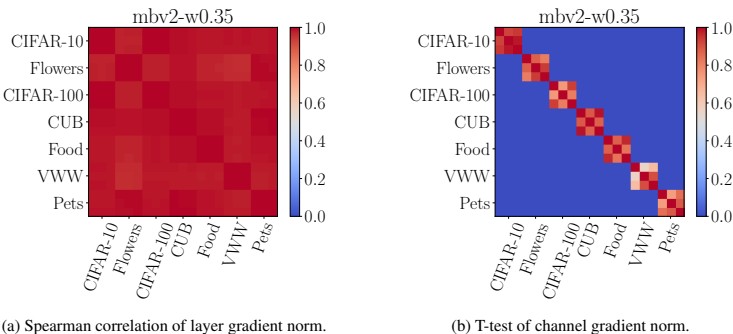

(a) Spearman correlation of layer gradient norm.

(b) T-test of channel gradient norm.

Figure 3: Validation of channel and layer proposition across seeds and datasets for MobileNetV2.

tasks. Consistently across all scenarios, we observe that $\alpha$ remains below two, confirming the heavy-tailed nature of stochastic gradients. Interestingly, MCUNet exhibits significantly more heavy-tailed gradient behavior compared to other architectures. We hypothesize this stems from its compressed architecture design, which renders it more parameter-efficient. This increased efficiency likely concentrates gradient information more densely in fewer parameters, intensifying the heavy-tailed characteristic of its gradient distribution.

**Layer Gradient Norm Distribution.** Next, we examine Proposition 3.1, which posits that the same network produces consistent layer gradient norm topologies across different downstream tasks. We define "layer topology" as a vector containing the cumulative gradient norm of each layer across all training epochs. To quantify similarity between topologies, we compute Spearman correlation coefficients between all possible pairs of fine-tuning runs across our seven downstream datasets, using three random seeds per dataset. This yields a comprehensive $21 \times 21$ correlation matrix for each network architecture, as visualized in Fig. 3a with MobileNetV2. The results provide strong empirical support for our proposition—even in the worst-case comparison between the most dissimilar dataset pairs, the correlation coefficient never falls below $0.8$. This remarkably high correlation confirms that layer-level gradient importance rankings remain largely invariant across diverse downstream tasks, validating our approach of pre-selecting layers based solely on architectural considerations.

**Channel Gradient Norm Distribution.** We now validate Proposition 3.2, which addresses gradient behavior at the channel level. Using methodology parallel to our layer analysis, we construct vectors where each element represents the cumulative gradient norm of a specific channel across all training epochs. This yields a high-dimensional representation of channel importance for each fine-tuning experiment. To assess whether these channel importance distributions differ significantly between tasks, we employ Student's T-test for pairwise comparisons, with results visualized in Fig. 3b. The analysis reveals a striking pattern: p-values for all inter-dataset comparisons are effectively zero, strongly rejecting the null hypothesis that channel gradient distributions from different datasets share the same mean values. This confirms our proposition that channel-level gradient importance patterns are fundamentally task-dependent and cannot be predetermined offline without access to the target dataset. Notably, the diagonal blocks in our visualization—representing comparisons between different random seeds for the same dataset—mostly feature non-zero p-values. This secondary finding indicates that while channel importance varies dramatically between tasks, it retains some

consistency across different initializations for the same task.

Results for other CNN architectures and transformers architectures show similar patterns (see Sec. D.2 and Sec. D.7 of the appendix).

## 4.2 MAIN RESULTS

**Preamble.** To rigorously validate the claims presented in Sec. 3, we systematically compare three distinct memory-constrained channel selection strategies. For each method, we explore both Static and Dynamic variations. In the Static approach, channel selection occurs once at initialization, with the same channels updated throughout training. The Dynamic approach reapplies the selection rule after each epoch, resulting in different channels being updated over time.

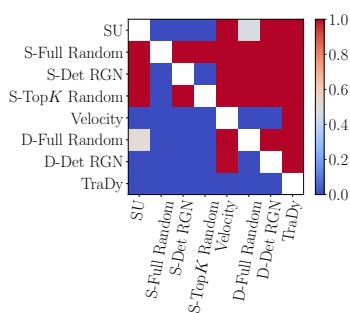

① *Full Random.* This baseline strategy randomly selects channels from throughout the entire network architecture, without layer-based prioritization. It serves as a control to evaluate the benefit of our layer selection approach.

② *Det RGN*: For each training epoch, we first compute the full gradient without updating network weights, then determin- istically select channels with the highest RGN values. While computationally impractical for real-world deployment (as it requires calculating the complete gradient), we expect this oracle-like method to serve as an upper-bound reference for performance.

Figure 4: T-test comparisons of average final test accuracies across multiple experimental di- mensions.

③ *TopK Random*: Randomly samples channels from within the predetermined subset of top $K$ layers. The practical choice of $K$ is defined in the appendix. In its dynamic version, this corresponds to our proposed algorithm TraDy.

We benchmark our method against Lin *et al.*'s Sparse Update (SU) scheme, which represents the current state-of-the-art in static channel selection for memory-constrained fine-tuning, as well as Quélennec *et al.*'s Velocity method, which dynamically selects neurons based on their output changes between epochs.

**Discussion.** In Fig. 4, we represent the results of paired t-tests comparing the average final test top-1 accuracies across all experimental conditions. Each cell represents a statistical comparison testing the hypothesis that the selection strategy on the y-axis achieves higher mean test accuracy than the strategy on the x-axis. We provide the complete table of results (Tab. 1 and Tab. 4) along with similar results for transformer architectures (Sec.D.7) and comparisons with full fine-tuning in the appendix. Regarding our introduced strategies, we observe that each dynamic variant (prefixed with D) outperforms its static counterpart (prefixed with S). Notably, while S-Full Random yields the worst results, our proposed algorithm—which restricts selection to top $K$ layers and incorporates dynamic selection—achieves the best performance, even surpassing D-RGN Deterministic, which was expected to serve as an upper bound. Velocity achieves the second-best accuracy performance among all evaluated methods, demonstrating the effectiveness of dynamic selection approaches.

We hypothesize that under extremely constrained memory budgets, D-RGN Deterministic's approach of always selecting channels with maximal RGN effectively leaves many channels with smaller but significant RGN values permanently frozen. This likely causes the training process to follow the direction of maximal gradient slope, potentially leading to local minima. In contrast, TraDy follows, on average, the same direction as the non-null gradient while introducing beneficial stochasticity, as layers with negligible gradients are excluded, but dynamic resampling occurs among significant ones. This hypothesis is further supported by S-TopK Random's poor performance (second worst strategy), highlighting that dynamic reselection is crucial for achieving good results.

**Efficiency Metrics Analysis.** Fig. 5 illustrates the temporal evolution of key efficiency metrics during MobileNetV2 fine-tuning on the Food dataset under the most restrictive memory constraint. These results showcase patterns that remain consistent across different network-dataset-budget combinations, with complete training metrics available in the following GitHub repository.

We observe that all of our methods achieve similar levels of weight and activation sparsity, respectively in the range of $93\%$ to $99\%$ and $97.5\%$ to $99.5\%$. SU trades off extremely low activation memory for higher weight memory, possibly linked to the evolutionary search process implicitly maximizing the amount of parameters updated. In comparison, we observe a more balanced trade-off

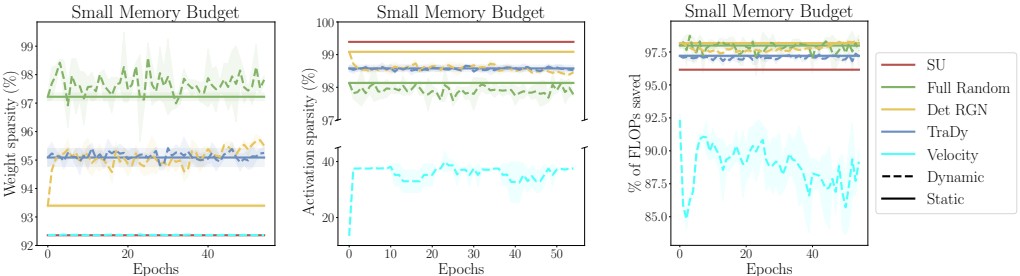

(a) Weight sparsity evolution during training.

(b) Activation sparsity evolution during training.

(c) Computational savings in weight derivative FLOPs.

Figure 5: Efficiency metrics comparison across channel selection strategies during MobileNetV2 fine-tuning on Food dataset under memory constraint. Results show evolution of sparsity levels and computational savings throughout training.

with our TraDy algorithm, suggesting that maximizing weight update does not necessarily result in improved task performance.

Notably, while Velocity achieves competitive accuracy (second-best among all methods), its neuron-level selection strategy results in substantially lower activation sparsity (20-40% range) compared to our channel-based approaches (97-99% range). This is because updating even a single neuron within a layer requires storing the complete activation map from the previous layer. Consequently, Velocity also achieves lower FLOPs savings (approximately 88%) compared to TraDy and other channel-based methods (97%). Moreover, due to its reweighting focusing solely on weight memory, the Velocity selection strategy results in the selection of computationally expensive neurons to update, further increasing FLOPs requirements relative to the other strategies explored in our paper. This is without accounting for the additional computational overhead of computing the velocity metric for all neurons or the memory overhead of storing all activation values necessary to compute this metric.

TraDy consistently requires significantly fewer FLOPs for weight derivative computation compared to both SU and Velocity. This computational advantage stems from the emergence of depthwise convolution layers among the top-ranked layers, which by design have low computational costs during both forward and backward passes.

The combination of both high weight and activation sparsity levels makes our method intrinsically more competitive than strategies that focus on either dimension alone. Methods like Jiang et al. (2022)'s Back Razor or Nguyen et al. (2025)'s ASI achieve similar levels of activation sparsity or compression rates but without the weight sparsity, while Velocity optimizes weight memory at the expense of activation memory. TraDy's balanced approach to both dimensions is particularly advantageous for on-device learning scenarios where all memory resources are strictly constrained.

## 5 CONCLUSION

In this work, we introduced TraDy, a memory-efficient transfer learning approach that dynamically selects channel subsets for update under tight resource constraints. Our method builds on two key insights: stochastic gradients often exhibit heavy-tailed behavior, leading to inherent sparsity, and layer importance remains consistent across tasks while channel relevance varies. By stochastically resampling channels between epochs within architecturally important layers, our approach proves its effectiveness in several challenging transfer learning scenarios, including training on efficient architectures designed for on-device deployment.

Future work will explore connections between stochastic channel selection and optimization theory, and extend our approach to broader network architectures for efficient on-device learning.

A CKNOWLEDGEMENTS

Part of this work was funded by the European Union's HORIZON Research and Innovation Programme under grant agreement No 101120657, project ENFIELD (European Lighthouse to Manifest Trustworthy and Green AI), by the European Union's Horizon Europe Research and Innovation Programme under grant agreement No. 101120237 (ELIAS), and by French National Research Agency (ANR-22-PEFT-0003 and ANR-22-PEFT-0007) as part of France 2030, the NF-NAI project and NF-FITNESS project. This work was also supported by the French National Research Agency (ANR) in the framework of the IA Cluster project "Hi! PARIS Cluster 2030" under Grant ANR-23-IACL-005, and by the Hi! PARIS Center on Data Analytics and Artificial Intelligence. Computation was performed thanks to GENCI-IDRIS ressources (Grant 2024-AD011014275R2.)

R EFERENCES

Toru Baji. Evolution of the gpu device widely used in ai and massive parallel processing. In *2018 IEEE 2nd Electron devices technology and manufacturing conference (EDTM)*, pp. 7–9. IEEE, 2018.

Lukas Bossard, Matthieu Guillaumin, and Luc Van Gool. Food-101–mining discriminative components with random forests. In *Computer Vision–ECCV 2014: 13th European Conference, Zurich, Switzerland, September 6-12, 2014, Proceedings, Part VI 13*, 2014.

Andrea Bragagnolo, Enzo Tartaglione, and Marco Grangetto. To update or not to update? neurons at equilibrium in deep models. *Advances in Neural Information Processing Systems*, 35, 2022.

Han Cai, Ligeng Zhu, and Song Han. ProxylessNAS: Direct neural architecture search on target task and hardware. In *International Conference on Learning Representations*, 2019. URL https://arxiv.org/pdf/1812.00332.pdf.

Han Cai, Chuang Gan, Ligeng Zhu, and Song Han. Tinytl: Reduce memory, not parameters for efficient on-device learning. *Advances in Neural Information Processing Systems*, 33:11285–11297, 2020.

Yu Cheng, Duo Wang, Pan Zhou, and Tao Zhang. Model compression and acceleration for deep neural networks: The principles, progress, and challenges. *IEEE Signal Processing Magazine*, 35 (1):126–136, 2018.

Aakanksha Chowdhery, Pete Warden, Jonathon Shlens, Andrew Howard, and Rocky Rhodes. Visual wake words dataset. *arXiv preprint arXiv:1906.05721*, 2019.

Dorottya Demszky, Kelvin Guu, and Percy Liang. Transforming question answering datasets into natural language inference datasets. *arXiv preprint arXiv:1809.02922*, 2018.

Jia Deng, Wei Dong, Richard Socher, Li-Jia Li, Kai Li, and Li Fei-Fei. Imagenet: A large-scale hierarchical image database. In *2009 IEEE conference on computer vision and pattern recognition*, 2009.

Lei Deng, Guoqi Li, Song Han, Luping Shi, and Yuan Xie. Model compression and hardware acceleration for neural networks: A comprehensive survey. *Proceedings of the IEEE*, 108(4): 485–532, 2020.

Li Deng, Geoffrey Hinton, and Brian Kingsbury. New types of deep neural network learning for speech recognition and related applications: An overview. In *2013 IEEE international conference on acoustics, speech and signal processing*, pp. 8599–8603. IEEE, 2013.

Priya Goyal, Piotr Dollár, Ross Girshick, Pieter Noordhuis, Lukasz Wesolowski, Aapo Kyrola, Andrew Tulloch, Yangqing Jia, and Kaiming He. Accurate, large minibatch sgd: Training imagenet in 1 hour. *arXiv preprint arXiv:1706.02677*, 2017.

Haoze He, Juncheng B Li, Xuan Jiang, and Heather Miller. Smt: Fine-tuning large language models with sparse matrices. In *The Thirteenth International Conference on Learning Representations*, 2025.

Geoffrey Hinton. The forward-forward algorithm: Some preliminary investigations. *arXiv preprint arXiv:2212.13345*, 2022.

Edward J Hu, Yelong Shen, Phillip Wallis, Zeyuan Allen-Zhu, Yuanzhi Li, Shean Wang, Lu Wang, Weizhu Chen, et al. Lora: Low-rank adaptation of large language models. *ICLR*, 1(2):3, 2022.

Ozlem Durmaz Incel and Sevda Ozge Bursa. On-device deep learning for mobile and wearable sensing applications: A review. *IEEE Sensors Journal*, 2023.

Ziyu Jiang, Xuxi Chen, Xueqin Huang, Xianzhi Du, Denny Zhou, and Zhangyang Wang. Back razor: Memory-efficient transfer learning by self-sparsified backpropagation. *Advances in neural information processing systems*, 35:29248–29261, 2022.

Gal Kaplun, Andrey Gurevich, Tal Swisa, Mazor David, Shai Shalev-Shwartz, and Eran Malach. Less is more: Selective layer finetuning with subtuning, 2023.

Jacob Devlin Ming-Wei Chang Kenton and Lee Kristina Toutanova. Bert: Pre-training of deep bidirectional transformers for language understanding, 2019.

Alex Krizhevsky, Geoffrey Hinton, et al. Learning multiple layers of features from tiny images, 2009.

Alex Krizhevsky, Ilya Sutskever, and Geoffrey E Hinton. Imagenet classification with deep convolutional neural networks. *Advances in neural information processing systems*, 25, 2012.

Young D. Kwon, Rui Li, Stylianos I. Venieris, Jagmohan Chauhan, Nicholas D. Lane, and Cecilia Mascolo. Tinytrain: Resource-aware task-adaptive sparse training of dnns at the data-scarce edge, 2024. URL https://arxiv.org/abs/2307.09988.

Yoonho Lee, Annie S Chen, Fahim Tajwar, Ananya Kumar, Huaxiu Yao, Percy Liang, and Chelsea Finn. Surgical fine-tuning improves adaptation to distribution shifts. *arXiv preprint arXiv:2210.11466*, 2022.

Ziyu Li, Enzo Tartaglione, and Van-Tam Nguyen. Scotti: Save computation at training time with an adaptive framework. In *Proceedings of the IEEE/CVF International Conference on Computer Vision*, pp. 1443–1452, 2023.

Ji Lin, Wei-Ming Chen, Yujun Lin, Chuang Gan, Song Han, et al. Mcunet: Tiny deep learning on iot devices. *Advances in Neural Information Processing Systems*, 33, 2020.

Ji Lin, Ligeng Zhu, Wei-Ming Chen, Wei-Chen Wang, Chuang Gan, and Song Han. On-device training under 256kb memory. *Advances in Neural Information Processing Systems*, 35, 2022.

Shih-Yang Liu, Chien-Yi Wang, Hongxu Yin, Pavlo Molchanov, Yu-Chiang Frank Wang, Kwang-Ting Cheng, and Min-Hung Chen. Dora: Weight-decomposed low-rank adaptation. In *Forty-first International Conference on Machine Learning*, 2024.

Yinhan Liu. Roberta: A robustly optimized bert pretraining approach. *arXiv preprint arXiv:1907.11692*, 364, 2019.

Ze Liu, Yutong Lin, Yue Cao, Han Hu, Yixuan Wei, Zheng Zhang, Stephen Lin, and Baining Guo. Swin transformer: Hierarchical vision transformer using shifted windows. In *Proceedings of the IEEE/CVF international conference on computer vision*, pp. 10012–10022, 2021.

Bradley McDanel, Helia Dinh, and John Magallanes. Accelerating dnn training with structured data gradient pruning. In *2022 26th International Conference on Pattern Recognition (ICPR)*, pp. 2293–2299. IEEE, 2022.

Shervin Minaee, Yuri Boykov, Fatih Porikli, Antonio Plaza, Nasser Kehtarnavaz, and Demetri Terzopoulos. Image segmentation using deep learning: A survey. *IEEE transactions on pattern analysis and machine intelligence*, 44(7):3523–3542, 2021.

Mohammad Mohammadi, Adel Mohammadpour, and Hiroaki Ogata. On estimating the tail index and the spectral measure of multivariate $\alpha$-stable distributions. *Metrika*, 78(5):549–561, 2015.

Ali Bou Nassif, Ismail Shahin, Imtinan Attili, Mohammad Azzeh, and Khaled Shaalan. Speech recognition using deep neural networks: A systematic review. *IEEE access*, 7:19143–19165, 2019.

Le-Trung Nguyen, Aël Quélennec, Enzo Tartaglione, Samuel Tardieu, and Van-Tam Nguyen. Activation map compression through tensor decomposition for deep learning. *arXiv preprint arXiv:2411.06346*, 2024.

Le-Trung Nguyen, Aël Quélennec, Van-Tam Nguyen, and Enzo Tartaglione. Beyond low-rank decomposition: A shortcut approach for efficient on-device learning. *arXiv preprint arXiv:2505.05086*, 2025.

Maria-Elena Nilsback and Andrew Zisserman. Automated flower classification over a large number of classes. In *2008 Sixth Indian Conference on Computer Vision, Graphics & Image Processing*, pp. 722–729. IEEE, 2008.

Omkar M Parkhi, Andrea Vedaldi, Andrew Zisserman, and CV Jawahar. Cats and dogs. In *2012 IEEE conference on computer vision and pattern recognition*, pp. 3498–3505. IEEE, 2012.

Danilo Pietro Pau and Fabrizio Maria Aymone. Suitability of forward-forward and pepita learning to mlcommons-tiny benchmarks. In *2023 IEEE International Conference on Omni-layer Intelligent Systems (COINS)*, 2023.

Adam Poliak. A survey on recognizing textual entailment as an nlp evaluation. *arXiv preprint arXiv:2010.03061*, 2020.

Aël Quélennec, Enzo Tartaglione, Pavlo Mozharovskyi, and Van-Tam Nguyen. Towards on-device learning on the edge: Ways to select neurons to update under a budget constraint. In *Proceedings of the IEEE/CVF Winter Conference on Applications of Computer Vision*, pp. 685–694, 2024.

Berkman Sahiner, Weijie Chen, Ravi K Samala, and Nicholas Petrick. Data drift in medical machine learning: implications and potential remedies. *The British Journal of Radiology*, 96(1150): 20220878, 2023.

Mark Sandler, Andrew Howard, Menglong Zhu, Andrey Zhmoginov, and Liang-Chieh Chen. Mobilenetv2: Inverted residuals and linear bottlenecks. In *Proceedings of the IEEE conference on computer vision and pattern recognition*, pp. 4510–4520, 2018.

Jaime Sevilla, Lennart Heim, Anson Ho, Tamay Besiroglu, Marius Hobbhahn, and Pablo Villalobos. Compute trends across three eras of machine learning. In *2022 International Joint Conference on Neural Networks (IJCNN)*, pp. 1–8. IEEE, 2022.

Karen Simonyan and Andrew Zisserman. Very deep convolutional networks for large-scale image recognition. *arXiv preprint arXiv:1409.1556*, 2014.

Umut Simsekli, Mert Gürbüzbalaban, Thanh Huy Nguyen, Gaël Richard, and Levent Sagun. On the heavy-tailed theory of stochastic gradient descent for deep neural networks. *arXiv preprint arXiv:1912.00018*, 222, 2019.

Richard Socher, Alex Perelygin, Jean Wu, Jason Chuang, Christopher D. Manning, Andrew Ng, and Christopher Potts. Recursive deep models for semantic compositionality over a sentiment treebank. In *Proceedings of the 2013 Conference on Empirical Methods in Natural Language Processing*, pp. 1631–1642, Seattle, Washington, USA, October 2013. Association for Computational Linguistics. URL https://www.aclweb.org/anthology/D13-1170.

Ian Tenney, Dipanjan Das, and Ellie Pavlick. Bert rediscovers the classical nlp pipeline. *arXiv preprint arXiv:1905.05950*, 2019.

Ashish Vaswani, Noam Shazeer, Niki Parmar, Jakob Uszkoreit, Llion Jones, Aidan N Gomez, Łukasz Kaiser, and Illia Polosukhin. Attention is all you need. *Advances in neural information processing systems*, 30, 2017.

Yijun Wan, Melih Barsbey, Abdellatif Zaidi, and Umut Simsekli. Implicit compressibility of overparametrized neural networks trained with heavy-tailed sgd. *arXiv preprint arXiv:2306.08125*, 2023.

Peter Welinder, Steve Branson, Takeshi Mita, Catherine Wah, Florian Schroff, Serge Belongie, and Pietro Perona. Caltech-ucsd birds 200. Technical Report CNS-TR-201, Caltech, 2010. URL `/se3/wp-content/uploads/2014/09/WelinderEtal10_CUB-200.pdf`,`http://www.vision.caltech.edu/visipedia/CUB-200.html`.

Sunghyeon Woo, Sol Namkung, Sunwoo Lee, Inho Jeong, Beomseok Kim, and Dongsuk Jeon. Paca: Partial connection adaptation for efficient fine-tuning. *arXiv preprint arXiv:2503.01905*, 2025.

Junhuan Yang, Yi Sheng, Yuzhou Zhang, Weiwen Jiang, and Lei Yang. On-device unsupervised image segmentation. *arXiv preprint arXiv:2303.12753*, 2023a.

Yuedong Yang, Guihong Li, and Radu Marculescu. Efficient on-device training via gradient filtering. In *Proceedings of the IEEE/CVF Conference on Computer Vision and Pattern Recognition*, pp. 3811–3820, 2023b.

Xucheng Ye, Pengcheng Dai, Junyu Luo, Xin Guo, Yingjie Qi, Jianlei Yang, and Yiran Chen. Accelerating cnn training by pruning activation gradients. In *Computer Vision–ECCV 2020: 16th European Conference, Glasgow, UK, August 23–28, 2020, Proceedings, Part XXV 16*, pp. 322–338. Springer, 2020.

Chiyuan Zhang, Samy Bengio, and Yoram Singer. Are all layers created equal? *ArXiv*, 2019.

Jianyu Zhang and Léon Bottou. Fine-tuning with very large dropout, 2024.

Zhi Zhang, Qizhe Zhang, Zijun Gao, Renrui Zhang, Ekaterina Shutova, Shiji Zhou, and Shanghang Zhang. Gradient-based parameter selection for efficient fine-tuning, 2024. URL `https://arxiv.org/abs/2312.10136`.

TABLE OF CONTENTS

# A    Limitations

**Related Works.** In Sec. 2, we discuss three key approaches to efficient subnetwork selection for on-device learning, yet our experimental comparisons focus only on Lin *et al.*'s SU method. Quelennec *et al.*'s implementation excludes activation memory from their budget calculations, making direct comparisons methodologically inconsistent with our approach which accounts for both weight and activation memory. Similarly, while Kwon *et al.*'s work offers improvements upon SU, the absence of publicly available code at the time of our research prevented us from implementing and benchmarking against their method.

**On-device Implementation.** Although our work aims to enable efficient on-device learning, we do not present metrics on actual hardware performance (latency, energy consumption, etc.). This limitation stems from our method's reliance on dynamic channel reselection between epochs, which requires specialized implementation for efficient execution on edge devices. Our current implementation serves as a simulation to demonstrate the potential algorithmic benefits, but further engineering work is needed to translate these theoretical gains into optimized on-device performance.

**Backpropagation Cost.** In our work, we report the FLOPs gained regarding the computation of weight derivatives. We however acknowledge that total backpropagation cost includes both weight and activation derivative calculations. The latter depends on the deepest layer requiring updates, as gradients must propagate from the output through all intermediate layers. Our approach typically selects relevant layers at greater depths than SU schemes, potentially increasing overall backpropagation latency despite weight derivative savings. In future work, we plan to explore techniques for exploiting the natural sparsity in activation gradients to enable compressed backpropagation, which would allow efficient updating of deeper layers with minimal accuracy degradation and reduced computational overhead.

# B    Link with Parameter-Efficient Fine-Tuning

Parameter-Efficient Fine-Tuning (PEFT) methods have emerged as a popular approach for adapting pre-trained models to downstream tasks while minimizing trainable parameters. Prominent adapter-based methods like LoRA Hu et al. (2022) and DoRA Liu et al. (2024) introduce low-rank decomposition matrices in parallel with frozen pre-trained weights, achieving impressive parameter efficiency by updating less than 1% of total parameters. LoRA decomposes weight updates into low-rank matrices that are trained alongside frozen weights, while DoRA further decomposes weights into magnitude and direction components, applying low-rank adaptation only to the directional component. These methods have demonstrated effectiveness across diverse architectures and tasks, establishing PEFT as a standard paradigm for efficient model adaptation.

However, adapter-based PEFT methods are fundamentally incompatible with extreme memory-constrained scenarios. These approaches introduce parallel computation paths that require computing forward pass through both weights and adapters paths during inference (thus increasing computational cost), and storing the full activation maps to update the adapters modules during backpropagation. Additionally, adapter modules introduce parameter storage overhead during training, as each forward pass must execute through both the frozen backbone and adapter pathways. These limitations render such methods impractical for on-device learning where activation memory constitutes the primary bottleneck.

Among PEFT methods, PaCA Woo et al. (2025) represents the closest approach to our setting, as it addresses both parameter and activation memory by randomly selecting channels for update within existing layers rather than introducing adapters. However, PaCA performs uniform random selection across all network layers without considering layer-wise gradient importance or dynamic resampling across epochs. In our experimental framework, this approach corresponds directly to our S-Full Random baseline, which we demonstrate to be the worst-performing selection strategy (Fig. 4) in our ablation study. In practice PaCa outperforms adapter-based strategies and by transitivity, TraDy provides further improvements in performance due to its innovations.

# C    Layer Ranking Consistency Detailed Analysis

Let us consider the simple case of $R$ convolutional layers having the same size, intercepted by ReLU activations, where a skip connection re-injects the input of the first in the final output $\mathcal{Y}$, reading

$\mathcal{Y} = \mathcal{A}_{i+R-1} + \mathcal{A}_i = \mathcal{C}_{\mathcal{W}_{i+R-1}} \circ \cdots \circ \mathcal{C}_{\mathcal{W}_i}(\mathcal{A}_i) + \mathcal{A}_i$. We also note $\mathcal{Z}_i$ the i-th layer pre-activation, $\mathbf{1}$ the indicator function, and $\odot$ the Hadamard product operator. According to (2), the weights derivatives could be further expressed as:

$$\frac{\partial \mathcal{L}}{\partial \mathcal{W}_{i+R-1}} = \mathrm{conv}\left(\mathcal{A}_{i+R-1}, \left[\frac{\partial \mathcal{L}}{\partial \mathcal{Y}} \odot \mathbf{1}_{\mathcal{Z}_{i+R-1}>0}\right]\right). \tag{10}$$

$\forall k \in [i, i+R-2]$, we define $\mathcal{J}$ with the following recursive expression:

$$\begin{cases} \mathcal{J}(i+R-1) = \left[\frac{\partial \mathcal{L}}{\partial \mathcal{Y}} \odot \mathbf{1}_{\mathcal{Z}_{i+R-1}>0}\right] \\ \\ \mathcal{J}(k) = \left(\mathrm{conv}\left(\mathcal{J}(k+1), \mathcal{W}_{k+1}^\top\right) \odot \mathbf{1}_{\mathcal{Z}_k>0}\right) \end{cases}. \tag{11}$$

Subsequently, $\forall k \in [i, i+R-2]$,

$$\frac{\partial \mathcal{L}}{\partial \mathcal{W}_k} = \mathrm{conv}\left(\mathcal{A}_k, \mathcal{J}(k)\right). \tag{12}$$

Besides, the input derivative is written as:

$$\frac{\partial \mathcal{L}}{\partial \mathcal{A}_i} = \frac{\partial \mathcal{L}}{\partial \mathcal{Y}} \cdot \left[I + \frac{\partial(\mathcal{C}_{\mathcal{W}_{i+R-1}} \circ \cdots \circ \mathcal{C}_{\mathcal{W}_{i+1}} \circ \mathcal{C}_{\mathcal{W}_i})}{\partial \mathcal{A}_i}\right], \tag{13}$$

with $I$ being the identity tensor.

Given that we're working with pre-trained networks, we can leverage specific properties established during their initial training phase. Pre-trained deep neural networks typically undergo regularization via weight decay and gradient clipping, which constrains weight norms to generally remain below one. Simultaneously, the inclusion of batch normalization layers during pre-training ensures that activation norms are similarly bounded. When fine-tuning on downstream tasks that share reasonable similarity with the pre-training domain, these weight and activation properties tend to be preserved, as the magnitude of weight adjustments remains relatively small.

For instance, when $\|\mathcal{W}_{k+1}\|_2 \leq 1$, $\forall k \in [i, i+R-2]$ and $\|\mathcal{A}_i\|_2 \leq 1$, we have that

$$\left\|\frac{\partial \mathcal{L}}{\partial \mathcal{W}_i}\right\|_2 \leq \left\|\frac{\partial \mathcal{L}}{\partial \mathcal{W}_{i+1}}\right\|_2 \leq \cdots \leq \left\|\frac{\partial \mathcal{L}}{\partial \mathcal{W}_{i+R-1}}\right\|_2. \tag{14}$$

While exceptions may occur—certain layers occasionally exhibit weight or activation norms exceeding one—these instances reflect inherent properties of the pre-trained network rather than task-specific adaptations. The fundamental insight is that layers consistently maintain their relative gradient norm proportions across diverse downstream tasks. This pattern becomes even more pronounced when using our reweighted gradient norm metric, as both channel weight and activation memory costs are architecture-dependent constants within each layer.

## D ADDITIONAL EXPERIMENTAL RESULTS

### D.1 EXPERIMENTAL SETUP

**Training.** Following Lin et al. (2022), we employ the same architectures pre-trained on ImageNet Deng et al. (2009): MobileNetV2 Sandler et al. (2018), ProxylessNAS Cai et al. (2019), and MCUNet Lin et al. (2020) (we load the weights provided in their code implementation). We perform training on a Nvidia Tesla V100 SXM2 and systematically train the classifier layer, independently of the freezing strategy. Algorithms are implemented in Python using PyTorch 2.0.0. We also provide results on transformer architectures in the appendix.

**Datasets.** We collect channel freezing metrics and transfer learning accuracy on multiple downstream datasets: CIFAR-10 Krizhevsky et al. (2009), CIFAR-100 Krizhevsky et al. (2009), CUB Welinder et al. (2010), Flowers Nilsback & Zisserman (2008), Food Bossard et al. (2014), Pets Parkhi et al. (2012) and VWW Chowdhery et al. (2019).[2] The learning policy consists of cosine learning rate decay with 5 warm-up epochs Goyal et al. (2017), 50 epochs for larger datasets (CIFAR-100, Food,

---

[2]Pets: https://creativecommons.org/licenses/by-sa/4.0/, CC BY-SA 4.0 license; ImageNet: https://image-net.org/download.php the ImageNet license; others are not listed.

and VWW), 100 epochs for CIFAR-10, and 200 epochs for smaller datasets (CUB, Pets, and Flowers). Learning rates range from $0.125$ to $0$, and we do not use weight decay or dropout. Our optimizer is Stochastic Gradient Descent (SGD) with no momentum (as keeping states in memory would go against our memory constraint). Each training is performed over three random seeds, and we report average results with standard deviation.

**Experiment Design.** For experimental consistency, we adopt the same memory budgets $B_{\text{mem}}$ used in the original SU work, implementing three distinct memory constraint levels for each network architecture. These budgets represent the maximum allowable memory consumption for both parameter storage and activations during the update process. Note that Velocity considers only weight memory in its budget calculations, so we evaluate it taking the weight component of the budget (excluding the activation part) for fair comparison. This comparative framework allows us to assess whether our theoretical insights translate into practical performance advantages while maintaining strictly equivalent memory constraints. For each channel selection strategy, we perform experiments on the cross-product of three networks, seven datasets, three memory budgets, and three seeds, producing 189 individual trainings per strategy, thus ensuring the statistical significance of the obtained results.

## D.2 PROPOSITIONS VALIDATION ON CNN ARCHITECTURES

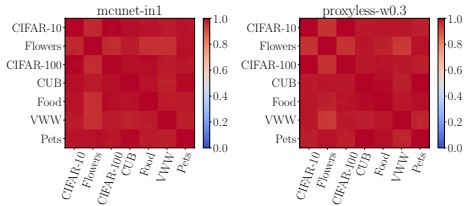 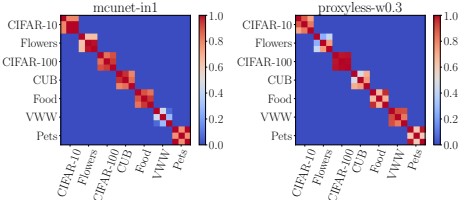

Figure 6: Spearman correlation of layer gradient norm across seeds and datasets.

Figure 7: T-test of channel gradient norm across seeds and datasets.

Fig. 6 and Fig. 7 respectively correspond to the validation of Proposition 3.1 (layer gradient norm consistency) and Proposition 3.2 (channel gradient norm variability) as presented in Sec. 4.1, with MCUNet and ProxylessNAS architectures. In both cases, the observed results are very similar to those obtained with MobileNetV2 (Fig. 3), thus providing further confirmation of the theoretical insights proposed in Sec. 3.

## D.3 REWEIGHTED GRADIENT NORM METRIC VALIDATION

Here, we experimentally validate the efficacy of our RGN metric introduced in Sec. 3.2. We conduct a series of experiments where channels are selectively frozen during training based on whether their gradient norm (either raw or reweighted) falls below a predefined threshold $\varepsilon$. Even though memory is not a channel freezing criterion in this setup, by logging metrics such as final accuracy, per-epoch memory, and FLOPs, we can observe how well a network converges given different levels of partial freezing. We consider a pre-trained MobileNetV2 that we fine-tune on CIFAR-10 and Flowers. The results of this study are shown in Fig. 8.

Each plot in Fig. 8 illustrates a progression of freezing strategies: the top-right corner represents a fully permissive threshold where all channels remain active during fine-tuning, while the bottom-left represents the most restrictive case where all channels (except the classifier) are frozen. Moving from right to left along each curve corresponds to increasingly stringent thresholds that progressively freeze more channels.

A key observation is that gradient norms naturally decrease during training, causing a fixed threshold $\varepsilon$ to freeze an increasing number of channels as training progresses. To capture this dynamic behavior across the entire training process, we present cumulative metrics for channel updates.

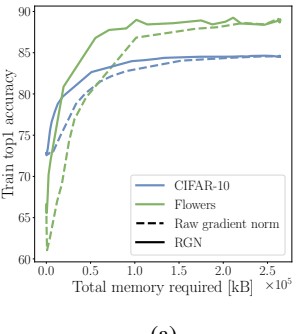 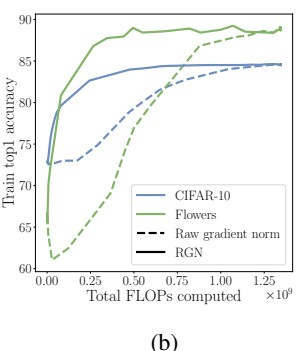 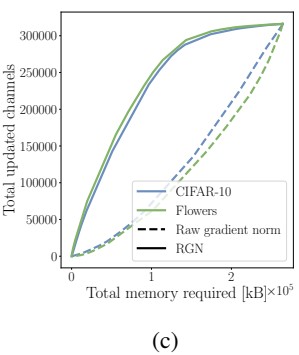

(a)    (b)    (c)

Figure 8: Channel thresholding results based on gradient norm. Each point represents a complete training run with respect to a pre-defined threshold $\varepsilon$. Plots show: (a) final accuracy vs. total memory usage, (b) final accuracy vs. total computational cost, and (c) total updated channel count vs. total memory usage.

Fig. 8a reveals significant difference between raw and reweighted norm-based pruning: with raw gradient thresholding, accuracy begins to deteriorate as soon as any memory reduction occurs. In contrast, the reweighted approach maintains full accuracy even when eliminating over half the total training memory. Fig. 8b shows analogous patterns for computational savings.

Fig. 8c helps us understand such phenomenon as, for the same amount of total memory, raw norm thresholding removes substantially more channels than the reweighted approach. This occurs because reweighting prioritizes freezing memory-intensive channels with relatively low gradient-to-memory ratios. In the raw scenario, channels with high gradient norms often coincide with high memory costs due to their larger parameter counts, yet these channels may have lower per-parameter importance. Our reweighting mechanism effectively identifies this inefficiency, allowing channels with high per-parameter gradient impact to be preserved while eliminating those with disproportionate memory requirements.

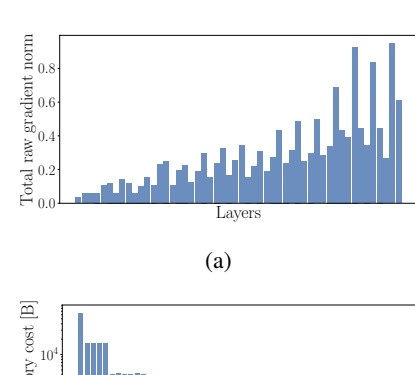

(a)

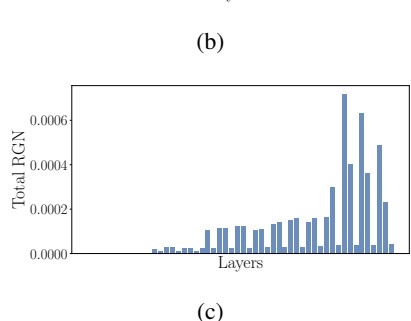

(b)

### D.4 LAYERS RGN BEHAVIOR

Fig. 9 illustrates how reweighting transforms the importance profile across network layers. Fig. 9a shows the raw gradient norm cumulated over training epochs, while Fig. 9c presents the corresponding reweighted values after accounting for channel memory costs (shown on a logarithmic scale in Fig. 9b). In Fig. 9c, we observe that some layers stand out in terms of cumulated RGN compared to others, namely the depthwise and the second point-wise layers of the blocks closer to the output.

With this observation and the knowledge that such topology is shared between downstream tasks and over time, we deduce that we can freeze a-priori a certain subset of layers as they provide a negligible contribution to the con-

(c)

Figure 9: Layers gradient norm and reweighting analysis in the case of a MobileNetV2 fine-tuned on CIFAR-10. Fig. 9a represents the raw cumulated gradient norm of layers, Fig. 9b the per-layer channel memory cost, and Fig. 9c the cumulated RGN of layers.

vergence of the network. To illustrate this point, we plot in Fig. 10 the evolution of cumulative RGN (expressed as a percentage of total network RGN) with respect to the number of layers considered (layers ranked in descending order of RGN).

We observe that for each network, half of the total network RGN is contained within less than a quarter of the layers and half of the layers correspond to more than 90% of the total RGN.

## D.5 TopK Layers Selection

Our proposed TraDy algorithm requires pre-selecting a subset of relevant layers for channel sampling. As established in Sec. 3.3 and experimentally confirmed in Fig. 6, the relative ranking of layers according to our RGN metric remains consistent across downstream tasks. This enables offline determination of layer importance by fine-tuning the target network on any available relevant downstream task and recording RGN values during training (even a few epochs suffice to establish reliable rankings).

The critical question becomes determining the optimal number $K$ of top-ranked layers to include in our selection pool. To investigate this parameter's impact, we conduct an experimental study using transfer learning with the smallest memory budget $B_{mem}$ on CIFAR-10. We use the gradients norm information to rank a-priori the layers along their total RGN and select different levels of top $K$ layers to perform sampling within ($K$ being the variable denoting the number of layers considered). We compare three dynamic channel selection strategies within the selected layer subsets:

1. Random selection of channels until memory budget $B_{mem}$ is met (Random).

2. Deterministic selection of channels with highest RGN values using complete gradient knowledge until $B_{mem}$ is met (Det RGN).

3. Deterministic selection based on raw gradient norm values using complete gradient knowledge until $B_{mem}$ is met (Det Raw Norm).

Fig. 11 presents the results of this analysis. For random selection, progressively excluding the least important layers initially improves training accuracy by constraining the sampling pool to more relevant channels. Performance peaks in the range around 35 to 40 layers before declining as essential layers are eliminated, highlighting their critical role in convergence. Notably, these top 35 layers capture 97% of the network's total RGN as can be observed in Fig. 10. We apply this 97% criterion to our other architectures, yielding 27 layers for MCUNet and 43 layers for ProxylessNAS.

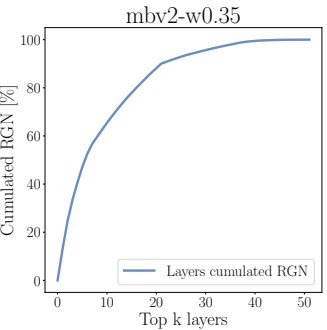

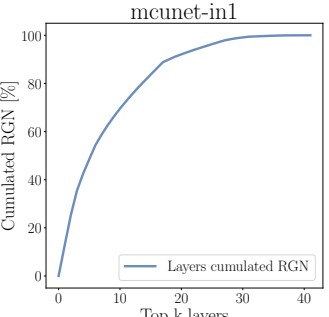

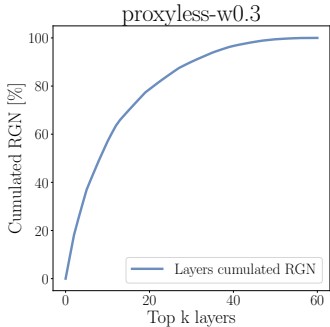

Figure 10: Layers RGN cumulative contribution for 3 networks, fine-tuning on CIFAR-10.

We acknowledge that our layer selection approach is relatively straightforward and presents opportunities for further refinement. For instance, $K$ could be adaptively determined based on the available memory budget, maintaining a constant ratio between the budget and the total memory requirements of the selected layers. However, developing such an adaptive scheme would require substantial theoretical analysis and extensive empirical validation, which is outside of the scope of this work and will be explored in future research. As demonstrated in Fig. 5, our current fixed-threshold approach for determining $K$ enables TraDy to achieve competitive performance against alternative strategies within the scope of this work.

The RGN-based deterministic approach exhibits remarkable stability across different values of $K$, maintaining consistent performance until a decline occurs when reducing from the top 10 to top 5 layers. This behavior aligns with our expectation that gradient importance concentrates heavily within a small subset of layers as expressed in Sec. 3.2.

The Raw Norm approach demonstrates more complex dynamics. When applied to the entire network, it yields relatively modest performance, but shows substantial improvement as the least important

layers are progressively excluded. This pattern suggests that these lower-ranked layers contain channels with high absolute gradient magnitudes but poor gradient-to-memory efficiency ratios, which our reweighting scheme effectively identifies and deprioritizes.

Intriguingly, Raw Norm selection achieves superior accuracy compared to either RGN or Random selection within the 20-40 layer range, potentially indicating alternative ways to balance gradient magnitude and memory efficiency beyond our current formulation. However, as stated in Sec. 3.4, accessing the gradient norm to perform channel selection is energy inefficient compared to a Random selection approach. Beyond the 20-layer threshold, both methods converge toward similar performance levels, likely because the reweighting has diminishing impact on channel ordering when focusing on the most gradient-rich layers where memory costs become more uniform.

While Lin et al. (2022) observed that depthwise layers contribute minimally to accuracy when updated in isolation, Zhang et al. (2019) demonstrated that layer contributions cannot be evaluated independently as they depend critically on which other layers remain frozen or active. Our findings suggest that the coordinated updating of depthwise layers alongside their corresponding second pointwise layers within each block creates synergistic effects that promote efficient convergence. This hypothesis is supported by the consistently high ranking we observe for these layer combinations, indicating their collective importance for gradient-based optimization under memory constraints.

## D.6 EXTENSION OF MAIN RESULTS

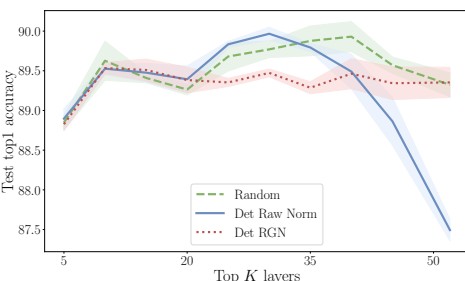

Figure 11: Final test top1 accuracies depending on the number of top $K$ layers for different dynamic channel selection strategies.

Here we provide additional results regarding our main experimental setup described in Sec. D.6. Tab. 1 presents a report of final test top-1 accuracies across our full experimental matrix spanning multiple architectures, datasets, and memory budgets when comparing SU and the dynamic selection strategies. The results for the static variants are displayed appart for readability in Sec. D.8 (Tab. 4).

We also provide comprehensive training metrics at the following GitHub repository in the *training_metrics* folder. This supplementary data includes detailed figures tracking multiple performance indicators across all fine-tuning experiments: training and test top-1 accuracies and losses, weight and activation sparsity percentages induced by each channel selection strategy, computational costs for weight derivative calculations (measured in FLOPs), relative FLOP savings compared to full fine-tuning, and additional memory-related control metrics. These extensive logs provide deeper insights into the behavior and efficiency characteristics of each evaluated approach. The anonymous repository also contains the complete source code required to reproduce our experimental results, accompanied by detailed execution instructions in the README file and a Jupyter notebook for generating all figures presented in this paper.

In Tab. 1, memory budgets $B_{\text{max}}$ are expressed as memory units, where each unit represents an individual memory slot. Actual memory consumption is calculated by multiplying these units by the number of bits per slot. The results reveal consistently low variance across repeated experiments for each combination of memory budget, architecture, dataset, and selection strategy. While accuracy differences between methods appear modest in individual comparisons, the extensive experimental validation across multiple dimensions provides strong statistical evidence for TraDy's superior performance. Additionally, TraDy offers practical advantages through its straightforward implementation compared to alternative approaches.

## D.7 RESULTS ON TRANSFORMERS ARCHITECTURES

Although TraDy was conceived with the goal of enabling on-device learning, it can easily be adapted to fine-tune larger architectures with limited memory or energy resources. In this section, we chose to consider a SwinT model Liu et al. (2021) pre-trained on ImageNet and fine-tuned on the seven downstream tasks introduced in the main paper. Regarding natural language processing (NLP), we consider both BERT Kenton & Toutanova (2019) and RoBERTa Liu (2019), standing as traditional

Table 1: Comparison of final top1 test accuracies between SU and dynamic channel selection strategies over various pretrained CNN models, datasets, and budgets.

| Model | $B_{mem}$ | Method | CIFAR-10 | CIFAR-100 | CUB | Flowers | Food | Pets | VWW | Average |
|---|---|---|---|---|---|---|---|---|---|---|
| MbV2-w0.35 | 27 946 | SU | 89.10±0.26 | 67.34±0.18 | 56.85±0.22 | 80.33±0.56 | 61.62±0.13 | 76.53±0.26 | 87.73±0.06 | 74.22±0.74 |
| | | Velocity | 89.84±0.19 | 68.14±0.17 | 57.51±0.30 | 79.46±0.32 | 61.79±0.12 | 76.24±0.23 | 88.29±0.18 | 74.47±0.60 |
| | | D-Full Random | 89.32±0.15 | 67.85±0.30 | 57.42±0.12 | 79.19±0.26 | 60.69±0.16 | 76.63±0.19 | 88.56±0.12 | 74.24±0.52 |
| | | D-Det RGN | 89.29±0.08 | 67.48±0.05 | 57.70±0.36 | 79.94±0.54 | 61.88±0.17 | 76.80±0.04 | 88.36±0.21 | 74.49±0.71 |
| | | TRaDy | 89.88±0.19 | 68.68±0.17 | 57.90±0.08 | 79.57±0.52 | 62.61±0.15 | 76.99±0.17 | 88.76±0.15 | **74.91±0.64** |
| | 66 592 | SU | 90.42±0.12 | 68.73±0.29 | 57.97±0.25 | 81.15±0.51 | 64.56±0.17 | 77.04±0.28 | 87.76±0.16 | 75.37±0.74 |
| | | Velocity | 90.94±0.31 | 69.74±0.45 | 58.45±0.59 | 80.13±0.62 | 65.43±0.23 | 77.14±0.37 | 88.31±0.22 | **75.73±1.13** |
| | | D-Full Random | 90.06±0.08 | 68.93±0.28 | 58.44±0.15 | 79.59±0.45 | 62.96±0.23 | 76.88±0.13 | 88.76±0.34 | 75.09±0.70 |
| | | D-Det RGN | 90.26±0.05 | 68.82±0.13 | 58.73±0.10 | 80.58±0.40 | 64.22±0.20 | 76.65±0.52 | 88.25±0.15 | 75.36±0.72 |
| | | TRaDy | 90.79±0.21 | 69.57±0.27 | 59.09±0.15 | 80.09±0.51 | 64.96±0.22 | 76.64±0.11 | 88.22±0.32 | 75.62±0.75 |
| | 93 696 | SU | 90.69±0.17 | 69.17±0.09 | 57.92±0.35 | 81.09±0.39 | 65.33±0.23 | 77.12±0.16 | 87.30±0.32 | 75.52±0.70 |
| | | Velocity | 91.37±0.19 | 70.62±0.13 | 59.03±0.29 | 80.57±0.26 | 66.69±0.3 | 76.67±0.33 | 88.11±0.3 | **76.15±0.70** |
| | | D-Full Random | 90.69±0.16 | 69.41±0.22 | 58.74±0.08 | 79.99±0.51 | 63.90±0.22 | 76.51±0.40 | 88.85±0.22 | 75.44±0.77 |
| | | D-Det RGN | 90.70±0.13 | 69.41±0.28 | 58.86±0.20 | 80.93±0.43 | 65.48±0.07 | 76.96±0.23 | 87.84±0.06 | 75.74±0.62 |
| | | TRaDy | 90.95±0.33 | 70.04±0.03 | 58.91±0.15 | 80.76±0.37 | 65.89±0.04 | 77.21±0.32 | 88.01±0.35 | 75.97±0.70 |
| | 1 252 320 | Baseline | 92.72±0.03 | 72.69±0.16 | 60.03±0.18 | 81.88±0.34 | 70.79±0.20 | 76.68±0.33 | 88.58±0.19 | 77.62±0.60 |
| MCUNet-in1 | 15 936 | SU | 89.51±0.23 | 68.41±0.27 | 60.68±0.27 | 82.92±0.43 | 65.57±0.06 | 81.15±0.29 | 89.14±0.10 | 76.77±0.69 |
| | | Velocity | 89.93±0.11 | 69.20±0.15 | 60.69±0.50 | 82.05±0.34 | 64.42±0.21 | 81.21±0.25 | 89.53±0.08 | 76.72±0.72 |
| | | D-Full Random | 90.22±0.06 | 69.08±0.24 | 61.21±0.22 | 82.37±0.26 | 65.71±0.16 | 81.20±0.16 | 89.96±0.05 | 77.11±0.48 |
| | | D-Det RGN | 90.29±0.2 | 69.06±0.28 | 61.03±0.40 | 82.34±0.37 | 65.95±0.07 | 81.07±0.13 | 89.90±0.19 | 77.09±0.69 |
| | | TRaDy | 90.38±0.18 | 69.72±0.14 | 61.30±0.20 | 82.54±0.59 | 66.78±0.17 | 81.10±0.11 | 89.79±0.27 | **77.37±0.74** |
| | 64 832 | SU | 91.65±0.26 | 70.96±0.23 | 62.03±0.32 | 83.79±0.53 | 69.77±0.03 | 81.52±0.11 | 88.67±0.14 | 78.34±0.73 |
| | | Velocity | 92.24±0.03 | 72.31±0.24 | 62.56±0.42 | 82.80±0.51 | 70.42±0.17 | 81.40±0.59 | 89.37±0.23 | **78.73±0.96** |
| | | D-Full Random | 91.70±0.13 | 71.58±0.18 | 62.43±0.10 | 82.33±0.31 | 69.07±0.28 | 81.26±0.09 | 89.75±0.16 | 78.30±0.52 |
| | | D-Det RGN | 91.60±0.19 | 71.11±0.15 | 61.86±0.36 | 82.99±0.56 | 69.53±0.19 | 80.97±0.92 | 89.32±0.04 | 78.20±1.18 |
| | | TRaDy | 92.16±0.25 | 72.11±0.40 | 62.20±0.10 | 83.02±0.52 | 70.57±0.17 | 81.11±0.28 | 89.30±0.27 | 78.64±0.83 |
| | 112 640 | SU | 92.07±0.13 | 71.58±0.15 | 61.44±0.41 | 83.74±0.47 | 71.02±0.15 | 81.07±0.24 | 88.77±0.31 | 78.53±0.78 |
| | | Velocity | 92.90±0.11 | 73.66±0.29 | 62.53±0.51 | 82.99±0.54 | 72.32±0.14 | 80.87±0.45 | 89.36±0.04 | **79.23±0.93** |
| | | D-Full Random | 92.20±0.18 | 72.71±0.16 | 62.85±0.11 | 82.84±0.03 | 70.70±0.05 | 81.30±0.07 | 89.54±0.17 | 78.88±0.33 |
| | | D-Det RGN | 92.01±0.03 | 72.30±0.13 | 62.36±0.56 | 83.02±0.37 | 71.16±0.32 | 80.76±0.27 | 89.15±0.17 | 78.68±0.82 |
| | | TRaDy | 92.53±0.21 | 72.95±0.27 | 62.12±0.14 | 83.25±0.36 | 71.88±0.12 | 81.29±0.25 | 89.39±0.31 | 79.06±0.66 |
| | 1 309 808 | Baseline | 93.87±0.10 | 76.03±0.18 | 61.62±0.62 | 83.45±0.42 | 75.74±0.14 | 79.49±0.60 | 90.06±0.16 | 80.04±1.00 |
| Proxyless-w0.3 | 25 984 | SU | 91.00±0.25 | 68.94±0.16 | 57.04±0.36 | 82.36±0.25 | 63.30±0.11 | 78.96±0.43 | 88.26±0.26 | 75.69±0.74 |
| | | Velocity | 90.69±0.10 | 69.12±0.06 | 55.98±0.12 | 81.85±0.34 | 61.46±0.13 | 78.58±0.50 | 88.67±0.20 | 75.19±0.67 |
| | | D-Full Random | 90.76±0.23 | 69.20±0.24 | 56.55±0.13 | 81.54±0.64 | 62.69±0.12 | 78.64±0.29 | 88.90±0.11 | 75.47±0.80 |
| | | D-Det RGN | 91.06±0.04 | 69.20±0.16 | 57.70±0.34 | 81.80±0.64 | 64.22±0.16 | 78.72±0.45 | 88.71±0.12 | 75.92±0.89 |
| | | TRaDy | 91.34±0.14 | 69.83±0.46 | 57.62±0.26 | 82.13±0.34 | 64.30±0.21 | 78.73±0.48 | 88.86±0.21 | **76.12±0.86** |
| | 72 960 | SU | 91.88±0.27 | 70.34±0.19 | 58.33±0.36 | 83.15±0.28 | 66.49±0.29 | 78.99±0.74 | 87.82±0.12 | 76.71±0.98 |
| | | Velocity | 92.37±0.03 | 71.84±0.08 | 58.72±0.72 | 82.35±0.20 | 66.63±0.13 | 78.9±0.44 | 88.53±0.18 | 77.05±0.90 |
| | | D-Full Random | 91.97±0.36 | 71.04±0.11 | 58.22±0.43 | 81.63±0.78 | 65.62±0.36 | 79.00±0.28 | 88.86±0.25 | 76.62±1.10 |
| | | D-Det RGN | 91.92±0.26 | 70.67±0.16 | 58.72±0.23 | 82.51±0.55 | 66.83±0.03 | 79.20±0.6 | 88.08±0.18 | 76.85±0.92 |
| | | TRaDy | 92.27±0.36 | 71.39±0.27 | 58.80±0.47 | 82.39±0.18 | 67.17±0.10 | 79.10±0.14 | 88.31±0.23 | **77.06±0.73** |
| | 101 376 | SU | 92.42±0.16 | 71.32±0.12 | 58.52±0.25 | 83.24±0.25 | 67.18±0.09 | 79.03±0.24 | 87.92±0.17 | 77.09±0.51 |
| | | Velocity | 92.82±0.23 | 72.51±0.22 | 59.71±0.46 | 82.61±0.46 | 68.18±0.06 | 79.01±0.22 | 88.40±0.15 | **77.61±0.77** |
| | | D-Full Random | 92.21±0.01 | 71.54±0.21 | 58.86±0.42 | 82.41±0.16 | 66.69±0.02 | 78.80±0.39 | 89.04±0.39 | 77.08±0.74 |
| | | D-Det RGN | 92.37±0.09 | 71.06±0.02 | 59.27±0.61 | 82.73±0.51 | 67.97±0.19 | 79.06±0.63 | 88.1±0.03 | 77.22±1.04 |
| | | TRaDy | 92.50±0.24 | 72.18±0.33 | 59.34±0.25 | 82.80±0.45 | 68.05±0.21 | 79.29±0.28 | 88.06±0.24 | 77.46±0.78 |
| | 1 162 032 | Baseline | 93.71±0.12 | 74.81±0.13 | 61.75±0.12 | 84.44±0.50 | 72.98±0.09 | 78.53±0.10 | 88.95±0.04 | 79.31±0.56 |



Figure 12: Spearman correlation of layer gradient norm across seeds and datasets.

Figure 13: T-test of channel gradient norm across seeds and datasets.

Table 2: Comparison of final top1 test accuracies between dynamic channel selection strategies with a pretrained SwinT model fine-tuned on various datasets and budgets.

| Model | $B_{mem}$ | Method | CIFAR-10 | CIFAR-100 | CUB | Flowers | Food | Pets | VWW | Average |
|---|---|---|---|---|---|---|---|---|---|---|
| SwinT | 27 946 | D-Full Random | 96.34±0.09 | 82.77±0.10 | 71.83±4.14 | 88.64±0.36 | 80.66±0.09 | 90.76±0.33 | 93.79±0.07 | 86.40±4.17 |
| | | D-Det RGN | 96.60±0.04 | 83.18±0.04 | 74.56±0.31 | 88.52±0.28 | 81.25±0.10 | 90.91±0.29 | 93.25±0.17 | **86.90±0.55** |
| | | TRaDy | 96.30±0.06 | 82.85±0.16 | 74.40±0.13 | 88.61±0.51 | 80.75±0.05 | 91.15±0.20 | 93.73±0.09 | 86.83±0.60 |
| | 112 640 | D-Full Random | 96.59±0.20 | 83.44±0.09 | 72.76±0.32 | 82.26±6.56 | 80.88±0.45 | 90.61±0.8 | 93.92±0.10 | 85.78±6.64 |
| | | D-Det RGN | 96.82±0.07 | 83.77±0.05 | 74.67±0.40 | 89.51±0.04 | 82.43±0.09 | 90.78±0.11 | 92.96±0.14 | **87.28±0.46** |
| | | TRaDy | 96.74±0.07 | 83.55±0.13 | 74.30±0.14 | 88.60±0.44 | 81.56±0.10 | 91.11±0.24 | 93.83±0.02 | 87.10±0.55 |
| | 633 859 | D-Full Random | 97.06±0.12 | 84.65±0.24 | 75.11±0.39 | 89.10±0.17 | 83.59±0.12 | 90.95±0.22 | 93.25±0.04 | 87.67±0.56 |
| | | D-Det RGN | 97.37±0.08 | 85.11±0.09 | 75.20±0.08 | 90.45±0.51 | 83.94±0.05 | 91.39±0.07 | 93.32±0.24 | **88.11±0.59** |
| | | TRaDy | 97.25±0.03 | 84.65±0.24 | 75.11±0.39 | 89.10±0.17 | 83.59±0.12 | 90.95±0.22 | 93.25±0.04 | 87.70±0.55 |
| | 2 767 686 | D-Full Random | 97.40±0.07 | 85.77±0.09 | 75.89±0.29 | 90.00±0.49 | 84.76±0.13 | 91.55±0.38 | 93.74±0.17 | 88.44±0.73 |
| | | D-Det RGN | 97.64±0.06 | 85.88±0.12 | 76.26±0.42 | 91.46±0.52 | 84.95±0.05 | 91.20±0.20 | 93.88±0.17 | **88.75±0.73** |
| | | TRaDy | 97.62±0.09 | 85.77±0.09 | 75.89±0.29 | 90.00±0.49 | 84.76±0.13 | 91.55±0.38 | 93.74±0.17 | 88.48±0.73 |
| | 31 889 952 | Baseline | 97.78±0.16 | 86.30±0.05 | 74.89±0.20 | 90.57±0.43 | 86.07±0.23 | 90.18±0.60 | 93.72±0.10 | 88.50±0.31 |

NLP architectures and which we fine-tune on three tasks: QNLI Demszky et al. (2018), RTE Poliak (2020) and SST2 Socher et al. (2013).

In Fig. 12 and Fig. 13, we reproduce for these three architectures, the layer gradient norm Spearman correlation and t-test of channel gradient norm as introduced in Sec. D.3. Regarding the Spearman correlation, Fig. 12 provides confirmation that the transformer architectures considered also follow the layer invariance with respect to downstream tasks as introduced in Proposition 3.1. Similarly, Fig. 13, showcases the rejection of the hypothesis of channel topology preservation between datasets. Interestingly, in the case of the simpler convolutional architectures, we observed in Fig. 7 that the different seeds of the same dataset resulted in a similar channel topology. In the case of the transformer architectures, we observe that in most cases, different seeds for the same dataset does not necessarily result in similar channel topology. We suppose that this is due to the higher expressivity of these complex architectures, allowing for different subnetworks to perform the same task.

Tab. 2 and Tab. 3 present test accuracies for SwinT and BERT-family models respectively, evaluated using our three dynamic selection approaches (static approaches result are provided separately in Sec. D.8 for improved readability). Given that these transformer architectures are substantially larger than the CNNs in our main experiments (approximately 26-30× larger for SwinT and 77-90× larger for BERT/RoBERTa in terms of combined weight and activation memory), we explore two distinct memory constraint scenarios:

1. *Absolute Budget Matching*: We apply identical memory budgets to those used for CNN experiments. For transformer architectures, these budgets represent dramatically smaller proportions of the total network, simulating extreme resource constraints where users seek to exploit large model capabilities with severely limited computational resources.

2. *Proportional Budget Matching*: We scale memory budgets to maintain equivalent proportions of total model memory as in the CNN experiments, enabling more substantial network portions to participate in updates during each epoch.

This dual-budget approach allows us to evaluate our method's effectiveness across different constraint severity levels while providing insights into transformer fine-tuning behavior under varying resource limitations.

When comparing SwinT against CNN architectures, all three channel selection methods achieve superior accuracy even under the most restrictive memory constraints (less than 0.1% of total network

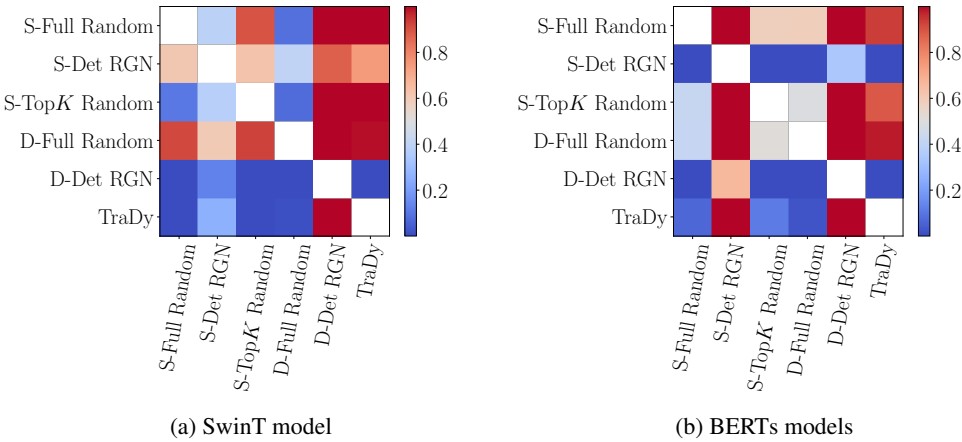



(a) SwinT model        (b) BERTs models



Figure 14: T-test comparisons of average final test accuracies across multiple experimental dimensions for each group of transformer architectures.

memory). Furthermore, both vision and NLP transformers exhibit smaller accuracy degradation between the most constrained budgets and full fine-tuning baselines compared to CNNs, despite these budgets constituting much smaller network fractions. This resilience underscores transformers' capacity to learn rich, transferable representations during pre-training that remain effective with minimal parameter updates during downstream adaptation.

Fig. 14 replicates the statistical analysis from Fig. 4 for transformer architectures. For SwinT (Fig. 14a), we observe an overall consistent strategy ranking with D-Det RGN achieving the best performance, followed by TraDy. However, for BERT-family models, both approaches appear to be outperformed by S-Det RGN, though this result carries greater uncertainty due to the smaller experimental sample size. Additionally, the substantial scale of these architectures suggests that our top $K$ layer selection methodology, while effective for CNNs, may require more sophisticated calibration for transformer models of this magnitude.

**Limitations and Future Directions for Transformer Architectures.** While our experimental results on transformer architectures demonstrate the applicability of TraDy's core principles, the performance gap compared to CNN architectures suggests

Table 3: Comparison of final top1 test accuracies between dynamic channel selection strategies with pretrained BERT and RoBERTa models, fine-tuned on various datasets and budgets.

| Model | $B_{\text{mem}}$ | Method | QNLI | RTE | SST2 | Average |
|---|---|---|---|---|---|---|
| BERT | 27 946 | D-Full Random | 84.50±0.23 | 56.32±0.36 | 89.41±0.46 | 76.74±0.63 |
| | | D-Det RGN | 87.78±0.45 | 57.28±1.78 | 91.17±0.40 | **78.74±1.88** |
| | | TRaDy | 84.38±0.21 | 57.76±0.96 | 89.53±0.13 | 77.22±0.57 |
| | 112 640 | D-Full Random | 84.50±0.04 | 58.24±2.32 | 89.41±0.53 | 77.38±2.38 |
| | | D-Det RGN | 89.00±0.22 | 60.05±2.66 | 91.25±0.57 | **80.10±2.73** |
| | | TRaDy | 84.56±0.28 | 57.88±0.91 | 89.60±0.26 | 77.35±0.57 |
| | 1 912 629 | D-Full Random | 85.85±0.47 | 54.99±0.91 | 89.76±0.48 | 76.87±1.13 |
| | | D-Det RGN | 89.83±0.09 | 60.53±0.55 | 91.48±0.24 | **80.61±0.61** |
| | | TRaDy | 85.84±0.30 | 56.68±0.72 | 90.10±0.35 | 77.54±0.49 |
| | 8 351 308 | D-Full Random | 88.68±0.14 | 58.24±1.46 | 89.60±0.46 | 78.84±1.54 |
| | | D-Det RGN | 90.47±0.16 | 60.17±3.07 | 91.67±0.52 | **80.77±3.12** |
| | | TRaDy | 88.97±0.20 | 57.16±1.50 | 90.86±0.18 | 79.00±0.88 |
| | 96 225 792 | Baseline | 90.81±0.27 | 62.45±1.81 | 91.74±0.50 | 81.67±1.90 |
| RoBERTa | 27 946 | D-Full Random | 89.69±0.04 | 57.40±0.72 | 93.23±0.34 | 80.11±0.80 |
| | | D-Det RGN | 90.97±0.22 | 76.29±0.55 | 92.51±0.40 | **86.59±0.71** |
| | | TRaDy | 89.71±0.13 | 57.16±0.75 | 93.31±0.07 | 80.06±0.76 |
| | 112 640 | D-Full Random | 89.99±0.26 | 58.12±1.25 | 93.12±0.20 | 80.41±1.29 |
| | | D-Det RGN | 90.78±0.23 | 77.02±0.21 | 93.00±0.11 | **86.93±0.33** |
| | | TRaDy | 90.05±0.12 | 59.57±2.87 | 93.31±0.13 | 80.98±1.66 |
| | 1 912 629 | D-Full Random | 91.23±0.18 | 65.10±0.83 | 93.85±0.65 | 83.39±1.07 |
| | | D-Det RGN | 91.23±0.26 | 75.09±2.25 | 93.04±0.26 | **86.45±2.28** |
| | | TRaDy | 91.23±0.26 | 68.83±1.78 | 93.43±1.16 | 84.50±1.24 |
| | 8 351 308 | D-Full Random | 91.54±0.11 | 73.77±2.92 | 93.27±0.13 | **86.19±2.92** |
| | | D-Det RGN | 91.90±0.13 | 70.28±15.25 | 93.58±0.11 | 85.25±15.25 |
| | | TRaDy | 91.36±0.23 | 73.53±1.78 | 93.31±0.35 | 86.07±1.83 |
| | 96 225 792 | Baseline | 92.31±0.14 | 76.41±0.55 | 93.16±0.92 | 87.29±1.08 |

that deeper understanding of fine-tuning dynamics in transformers is necessary to achieve optimal results. Recent work on sparse matrix fine-tuning for LLMs He et al. (2025) provides valuable insights that could inform more effective adaptations of our approach. Specifically, their analysis reveals that attention layer V vectors require the most trainable parameters while MLPs need minimal updates—insights that could guide more informed layer selection strategies for transformer

Table 4: Comparison of final top1 test accuracies between static channel selection strategies over various pretrained vision models, datasets, and budgets.

| Model | $B_{mem}$ | Method | CIFAR-10 | CIFAR-100 | CUB | Flowers | Food | Pets | VWW | Average |
|---|---|---|---|---|---|---|---|---|---|---|
| MbV2-w0.35 | 27 946 | S-Full Random | 87.79±0.21 | 66.52±0.07 | 55.6±0.55 | 79.15±0.43 | 58.21±0.21 | 76.46±0.04 | 88.17±0.03 | 73.13±0.76 |
| | | S-Det RGN | 88.78±0.07 | 67.25±0.12 | 57.11±0.49 | 79.59±0.41 | 60.05±0.11 | 76.70±0.26 | 88.44±0.15 | **73.99±0.73** |
| | | S-Top$K$ Random | 88.52±0.02 | 66.93±0.14 | 56.52±0.55 | 79.68±0.62 | 59.52±0.39 | 76.71±0.55 | 88.12±0.58 | 73.71±1.22 |
| | 66 592 | S-Full Random | 88.62±0.12 | 66.81±0.12 | 56.75±0.09 | 79.28±0.63 | 59.75±0.78 | 76.50±0.51 | 87.87±0.09 | 73.65±1.14 |
| | | S-Det RGN | 89.88±0.06 | 68.14±0.20 | 57.84±0.17 | 80.31±0.25 | 62.70±0.13 | 76.61±0.54 | 88.11±0.23 | **74.80±0.70** |
| | | S-Top$K$ Random | 89.76±0.27 | 68.10±0.18 | 57.49±0.36 | 80.39±0.46 | 62.08±0.29 | 76.64±0.24 | 87.75±0.13 | 74.60±0.78 |
| | 93 696 | S-Full Random | 89.01±0.22 | 67.21±0.15 | 56.87±0.51 | 79.78±0.79 | 60.40±0.56 | 76.58±0.35 | 88.22±0.48 | 74.01±1.27 |
| | | S-Det RGN | 90.45±0.10 | 68.84±0.04 | 57.80±0.09 | 80.59±0.11 | 64.11±0.19 | 76.75±0.26 | 87.54±0.06 | **75.15±0.37** |
| | | S-Top$K$ Random | 90.25±0.05 | 68.41±0.36 | 57.94±0.18 | 80.42±0.33 | 63.14±0.18 | 76.67±0.22 | 87.61±0.15 | 74.92±0.61 |
| MCUNet-in1 | 15 936 | S-Full Random | 88.78±0.17 | 67.78±0.35 | 60.14±0.18 | 82.20±0.45 | 62.68±0.38 | 81.09±0.19 | 89.50±0.23 | 76.02±0.79 |
| | | S-Det RGN | 89.12±0.14 | 67.97±0.10 | 60.06±0.22 | 82.14±0.37 | 63.77±0.15 | 80.79±0.34 | 89.55±0.01 | 76.20±0.59 |
| | | S-Top$K$ Random | 89.08±0.11 | 67.86±0.30 | 60.26±0.17 | 82.34±0.82 | 63.60±0.20 | 81.09±0.30 | 89.56±0.11 | **76.26±0.97** |
| | 64 832 | S-Full Random | 90.02±0.52 | 69.70±0.02 | 60.98±0.17 | 82.67±0.20 | 64.87±1.29 | 80.95±0.53 | 89.44±0.06 | 76.95±1.51 |
| | | S-Det RGN | 89.97±0.18 | 67.78±0.19 | 61.80±0.24 | 80.86±0.96 | 65.16±0.24 | 81.76±0.56 | 89.27±0.12 | 76.66±1.20 |
| | | S-Top$K$ Random | 90.82±0.27 | 70.34±0.15 | 60.90±0.14 | 82.92±0.54 | 67.29±0.28 | 81.34±0.29 | 89.11±0.06 | **77.53±0.76** |
| | 112 640 | S-Full Random | 90.82±0.13 | 70.75±0.41 | 61.22±0.11 | 82.77±0.35 | 66.92±0.77 | 80.80±0.15 | 89.01±0.12 | 77.47±0.97 |
| | | S-Det RGN | 91.28±0.13 | 71.53±0.20 | 61.02±0.16 | 82.69±0.58 | 69.17±0.23 | 80.64±0.16 | 88.89±0.17 | 77.89±0.73 |
| | | S-Top$K$ Random | 91.58±0.13 | 71.55±0.34 | 60.95±0.73 | 82.80±0.26 | 69.28±0.17 | 80.42±0.29 | 88.73±0.33 | **77.90±0.98** |
| Proxyless-w0.3 | 25 984 | S-Full Random | 89.15±0.33 | 67.90±0.21 | 55.22±0.23 | 81.64±0.54 | 58.72±0.26 | 78.32±0.14 | 88.39±0.08 | 74.19±0.77 |
| | | S-Det RGN | 90.19±0.27 | 68.50±0.20 | 57.13±0.25 | 81.89±0.37 | 61.69±0.19 | 78.90±0.14 | 88.51±0.10 | **75.26±0.61** |
| | | S-Top$K$ Random | 89.98±0.18 | 68.33±0.22 | 56.17±0.11 | 81.89±0.50 | 60.60±0.12 | 78.17±0.25 | 88.39±0.19 | 74.79±0.68 |
| | 72 960 | S-Full Random | 90.34±0.08 | 68.78±0.14 | 56.35±0.38 | 81.95±0.36 | 61.32±0.87 | 78.79±0.48 | 88.40±0.22 | 75.13±1.16 |
| | | S-Det RGN | 91.30±0.12 | 70.38±0.15 | 58.26±0.46 | 82.62±0.46 | 65.09±0.01 | 78.67±0.29 | 87.94±0.54 | **76.32±0.91** |
| | | S-Top$K$ Random | 91.09±0.14 | 70.10±0.32 | 57.32±0.45 | 82.15±0.10 | 63.75±0.16 | 78.4±0.32 | 87.86±0.40 | 75.81±0.79 |
| | 101 376 | S-Full Random | 90.64±0.15 | 69.37±0.14 | 57.18±0.63 | 82.15±0.33 | 62.60±0.63 | 78.51±0.20 | 88.06±0.32 | 75.50±1.04 |
| | | S-Det RGN | 91.76±0.15 | 71.28±0.35 | 58.6±0.18 | 82.82±0.42 | 66.45±0.25 | 78.70±0.49 | 87.84±0.27 | **76.78±0.85** |
| | | S-Top$K$ Random | 91.61±0.33 | 70.73±0.46 | 57.88±0.35 | 82.51±0.25 | 64.92±0.17 | 78.46±0.06 | 87.58±0.27 | 76.24±0.78 |
| SwinT | 27 946 | S-Full Random | 96.31±0.15 | 82.94±0.07 | 73.80±0.11 | 88.34±0.19 | 80.55±0.21 | 91.08±0.15 | 93.63±0.08 | 86.66±0.39 |
| | | S-Det RGN | 96.36±0.09 | 83.05±0.12 | 74.38±0.12 | 88.76±0.29 | 80.62±0.11 | 91.03±0.11 | 93.60±0.15 | **86.83±0.41** |
| | | S-Top$K$ Random | 96.28±0.07 | 82.97±0.09 | 74.03±0.07 | 88.58±0.11 | 80.58±0.25 | 90.87±0.21 | 93.70±0.04 | 86.72±0.37 |
| | 112 640 | S-Full Random | 96.54±0.10 | 83.25±0.36 | 74.00±0.23 | 88.74±0.09 | 81.09±0.07 | 91.21±0.11 | 93.64±0.11 | 86.92±0.48 |
| | | S-Det RGN | 96.70±0.06 | 83.59±0.16 | 74.68±0.20 | 89.36±0.32 | 81.97±0.09 | 90.97±0.24 | 92.99±0.06 | **87.18±0.49** |
| | | S-Top$K$ Random | 96.60±0.11 | 83.29±0.20 | 74.19±0.34 | 88.83±0.16 | 81.19±0.16 | 90.99±0.03 | 93.50±0.17 | 86.94±0.50 |
| | 633 859 | S-Full Random | 96.99±0.12 | 84.24±0.20 | 74.55±0.44 | 89.33±0.05 | 82.99±0.12 | 91.26±0.07 | 93.1±0.09 | 87.49±0.53 |
| | | S-Det RGN | 97.26±0.05 | 84.78±0.13 | 75.69±0.23 | 90.29±0.18 | 83.72±0.18 | 91.28±0.28 | 79.72±23.27 | 86.11±23.27 |
| | | S-Top$K$ Random | 97.06±0.06 | 84.24±0.20 | 74.55±0.44 | 89.33±0.05 | 82.99±0.12 | 91.26±0.07 | 93.10±0.09 | **87.50±0.52** |
| | 2 767 686 | S-Full Random | 97.50±0.06 | 85.53±0.08 | 75.75±0.14 | 89.84±0.08 | 84.46±0.12 | 91.30±0.23 | 93.73±0.17 | 88.30±0.36 |
| | | S-Det RGN | 97.50±0.06 | 85.53±0.08 | 75.75±0.14 | 89.84±0.08 | 84.46±0.12 | 91.30±0.23 | 93.73±0.17 | **88.30±0.36** |
| | | S-Top$K$ Random | 97.51±0.11 | 85.53±0.08 | 75.75±0.14 | 89.84±0.08 | 84.46±0.12 | 91.30±0.23 | 93.73±0.17 | 88.30±0.38 |

architectures. However, a key challenge remains: such analyses typically require a warm-up phase with full fine-tuning on downstream task data to compute Fisher information, whereas our method assumes no downstream task data is available a priori and enforces strict memory constraints at all times. Merging insights from layer-importance analysis in transformers with our dynamic channel selection framework represents a promising direction for future work, potentially enabling more effective memory-constrained fine-tuning of large language models.

## D.8    STATIC STRATEGIES RESULTS

This section presents experimental results for the static selection strategies evaluated in our study. Tab. 4 displays results for all vision architectures, while Tab. 5 presents findings for NLP models.

## E    LLM USAGE

The redaction of this paper received support from LLM to help improve grammar and readability. No scientific or technical content was generated through LLM. All numerical results, tables and figures are our own production.

Table 5: Comparison of final top1 test accuracies between static channel selection strategies with pretrained BERT and RoBERTa models, fine-tuned on various datasets and budgets.

| Model | $B_{\mathrm{mem}}$ | Method | QNLI | RTE | SST2 | Average |
|-------|------|--------|------|-----|------|---------|
| BERT | 27 946 | S-Full Random | 84.48±0.22 | 57.28±0.83 | 89.68±0.11 | 77.15±0.87 |
| | | S-Det RGN | 86.42±0.13 | 58.72±2.18 | 90.86±0.35 | **78.67±2.21** |
| | | S-Top$K$ Random | 84.53±0.34 | 58.24±2.32 | 89.37±0.13 | 77.38±2.35 |
| | 112 640 | S-Full Random | 84.51±0.08 | 58.72±2.40 | 89.72±0.48 | 77.65±2.45 |
| | | S-Det RGN | 88.30±0.43 | 59.09±1.46 | 91.21±0.46 | **79.53±1.59** |
| | | S-Top$K$ Random | 84.69±0.19 | 58.00±2.21 | 89.53±0.26 | 77.41±2.23 |
| | 1 912 629 | S-Full Random | 86.08±0.49 | 57.04±1.44 | 89.91±0.34 | 77.68±1.56 |
| | | S-Det RGN | 89.22±0.16 | 59.81±2.05 | 91.55±0.29 | **80.19±2.08** |
| | | S-Top$K$ Random | 86.80±0.43 | 58.60±2.05 | 90.29±0.07 | 78.56±2.10 |
| | 8 351 308 | S-Full Random | 88.55±0.17 | 56.80±0.21 | 90.29±0.18 | 78.55±0.32 |
| | | S-Det RGN | 89.87±0.22 | 61.01±2.53 | 91.55±0.29 | **80.81±2.57** |
| | | S-Top$K$ Random | 88.77±0.69 | 57.76±0.72 | 91.28±0.57 | 79.27±1.15 |
| RoBERTa | 27 946 | S-Full Random | 89.68±0.17 | 56.56±0.75 | 93.31±0.18 | 79.85±0.79 |
| | | S-Det RGN | 90.81±0.12 | 76.90±0.72 | 93.43±0.35 | **87.04±0.81** |
| | | S-Top$K$ Random | 89.66±0.07 | 56.68±0.72 | 93.31±0.07 | 79.88±0.73 |
| | 112 640 | S-Full Random | 89.69±0.10 | 59.09±3.62 | 93.39±0.29 | 80.72±3.63 |
| | | S-Det RGN | 90.91±0.61 | 76.77±2.61 | 92.85±0.35 | **86.85±2.70** |
| | | S-Top$K$ Random | 89.60±0.02 | 56.80±0.55 | 93.27±0.18 | 79.89±0.58 |
| | 1 912 629 | S-Full Random | 90.95±0.26 | 62.33±9.49 | 93.58±0.34 | 82.29±9.50 |
| | | S-Det RGN | 91.10±0.24 | 77.26±1.65 | 92.89±0.34 | **87.08±1.70** |
| | | S-Top$K$ Random | 91.14±0.22 | 62.33±6.14 | 93.23±0.46 | 82.23±6.16 |
| | 8 351 308 | S-Full Random | 91.28±0.26 | 72.80±0.75 | 92.78±0.00 | 85.62±0.79 |
| | | S-Det RGN | 91.20±0.31 | 75.09±0.00 | 93.20±0.66 | **86.49±0.73** |
| | | S-Top$K$ Random | 91.09±0.17 | 71.84±3.21 | 93.08±0.26 | 85.34±3.22 |

