# OpenReview forum: "Study of Training Dynamics for Memory-Constrained Fine-Tuning"
_ICLR.cc/2026/Conference — ICLR 2026 Poster_

### Official Review · Reviewer_LAwL · 2025-10-28

**Soundness:** 3
**Presentation:** 2
**Contribution:** 3
**Rating:** 6
**Confidence:** 3

**Summary:**

This paper proposes TraDy (Training Dynamics), a memory-efficient fine-tuning approach for deep neural networks under strict resource constraints. The method leverages two key insights: (1) layer importance for updates is architecture-dependent and can be determined a priori, and (2) dynamic stochastic channel selection provides superior gradient approximation compared to static approaches. The authors introduce a Reweighted Gradient Norm (RGN) metric that accounts for both gradient magnitude and memory costs. TraDy operates by pre-selecting important layers based on architectural properties, then dynamically resampling input channels between epochs within these layers. Experiments on CNNs and transformers across multiple vision and NLP tasks demonstrate that TraDy achieves competitive performance while maintaining up to 99% activation sparsity, 95% weight derivative sparsity, and 97% reduction in FLOPs for weight derivative computation.

**Strengths:**

1. The core idea is both creative and elegant. The central challenge of memory-constrained training is that one cannot afford to compute the very importance metrics needed to decide what to update. TraDy's solution—decoupling the problem into a static, a priori layer selection and a dynamic, stochastic channel selection—is a novel and highly effective way to break this circular dependency.

2. The authors compare TraDy against a comprehensive set of baselines, including static/dynamic and random/deterministic variants, across 3 CNN architectures, 7 vision datasets, 3 memory budgets, and 3 random seeds.

3. Well-structured paper with clear problem formulation and notation.

**Weaknesses:**

1. The main comparison is against Sparse Update (SU). More recent and relevant works are mentioned but not included in the main experiments. For example, how is the proposed method compared with "SMT: Fine-Tuing Large Language Models with Sparse Matrices".

2. LoRA and other parameter-efficient fine-tuning methods are not discussed or compared.

3. The authors acknowledge this limitation explicitly. The paper lacks actual on-device experiments showing latency, energy consumption, or throughput on target edge devices.

4. The paper is compactly formatted and the template style is changed.

**Questions:**

see weakness.

---

> ### Author Response · Authors · 2025-11-18
>
> ## W1: Comparison with Recent Works (SMT, LoRA, and Other PEFT Methods)
> Thank you for suggesting the comparison with "SMT: Fine-tuning Large Language Models with Sparse Matrices" (He et al.), which provides valuable insights regarding potential improvements of our method when adapting it to transformer architectures and larger models. In our response to Reviewer izYB's Q1, we provide a detailed analysis of the relationship between He et al.'s work and ours, highlighting key differences in our approach:
> - **No warm-up phase**: Unlike SMT, which requires full fine-tuning to compute importance metrics, our method operates under memory constraints at all times.
> - **Simpler optimizers**: We avoid memory-intensive optimizers like Adam to maintain extreme memory efficiency alongside the lack of momentum or weight decay terms.
>
> We believe that merging insights from SMT's layer-importance analysis for transformers with our dynamic channel selection represents a promising future direction. Additionally, we have provided a detailed comparison with Quélennec et al.'s Velocity strategy in our general response and revised submission, demonstrating TraDy's superior performance in both accuracy and activation sparsity.
>
> ## W2: LoRA and Parameter-Efficient Fine-Tuning Methods
> We address this important point in detail in our general response section of the rebuttal.
>
> ## W3: Lack of On-Device Experiments
> We fully acknowledge this limitation. As discussed in Appendix A (Limitations section), our current implementation serves as an algorithmic demonstration rather than an optimized on-device deployment. Translating our theoretical and simulated efficiency gains into actual hardware performance requires specialized engineering work. Our theoretical analysis in Sec. 3.4 shows that random channel selection has $\mathcal{O}(k\log(n))$ complexity which is negligible compared to gradient computation. Empirical validation on actual edge devices with measurements of latency, energy consumption, and throughput would indeed strengthen our claims. This represents important future work that we are committed to pursuing.
>
> ## W4: Paper Formatting and Template Style
> Thank you for this observation. Could you please provide more specific feedback regarding the formatting issues you noticed? In particular, it would be helpful if you could point to specific sections or elements of the manuscript that could be improved with better formatting. We are committed to addressing formatting concerns to meet the conference standards.

---

> > ### Comment · Reviewer_LAwL · 2025-11-26
> >
> > Regarding the template issue, it seems you removed the phrase "Under review at ICLR 2026" from the top margin.

---

> > > ### Author Response · Authors · 2025-11-26
> > >
> > > Thank you for pointing this out, we had not noticed the change from the template as it seems to be due to a LateX issue. We manually added it back to fully comply with the submission style.

---

### Official Review · Reviewer_FTbs · 2025-10-30

**Soundness:** 4
**Presentation:** 3
**Contribution:** 3
**Rating:** 8
**Confidence:** 3

**Summary:**

This paper propose a dynamic training method (TraDy) for memory-constrained scenarios. . By stochastically resampling channels between epochs within architecturally important layers, the experiments demonstrate the effectiveness of the proposed method across various transfer learning scenarios. This article is well-written and well-organized.

**Strengths:**

1. This paper is grounded in a solid theoretical foundation, featuring detailed derivations and rigorous argumentation.
2. This article is well-written and well-organized.

**Weaknesses:**

1.Fig. 2 is positioned too far from the corresponding text section. It is recommended to optimize the image layout.
2.The baseline for existing studies compared in this paper is limited, with only SU available.
3.It is recommended to highlight the best-performing results in Tables 1-5.

**Questions:**

1.Although the author claims that lennec et al.’s implementation excludes activation memory from their budget calculations, it doesn't seem to affect adding it as an additional baseline to Table 1-5. Why wasn't this done?

---

> ### Author Response · Authors · 2025-11-18
>
> ## W1: Figure 2 Positioning
> We understand how the positioning of Fig. 2 might be confusing in the original submission, and we have modified its placement in the revised manuscript to better align with the text layout. Our intention was to position it close to the heavy-tailed theory subsection (Section 3.2) to provide immediate visual evidence supporting our claim regarding the heavy-tailed behavior of stochastic gradients during fine-tuning.
>
> ## W2/Q1: Comparison with Quélennec et al.'s Velocity Method
> We initially excluded comparison with Quélennec et al.'s Velocity solution because their method does not account for activation memory but only weight memory, which we consider a critical limitation for memory-constrained on-device learning. However, as this concern was raised by multiple reviewers, we provide a detailed analysis of the comparison with TraDy in our general response and have updated the revised paper accordingly. Our results demonstrate that Velocity yields slightly worse accuracy than TraDy while achieving low activation sparsity levels and higher FLOPs demand, due to its output channel-focused selection strategy. This is a fundamental difference: output channel selection reduces weight memory but provides no activation memory savings, whereas our input channel selection simultaneously achieves both weight and activation sparsity. Given that activation memory often dominates the memory footprint during backpropagation, this distinction is crucial for truly memory-constrained scenarios. We have updated Tables 1-5 in the revised manuscript to highlight the best-performing results, which significantly improves readability and makes it easier to identify TraDy's advantages at a glance.

---

### Official Review · Reviewer_eoHC · 2025-10-31

**Soundness:** 3
**Presentation:** 2
**Contribution:** 2
**Rating:** 4
**Confidence:** 3

**Summary:**

The paper presents TraDy, a novel approach for fine-tuning pre-trained deep neural networks to downstream tasks under memory-constrained budgets. TraDy proposes a dynamic channel selection scheme that exploits the gradient sparsity of input channels to achieve both weight and activation sparsity during fine-tuning for downstream tasks.

**Strengths:**

- The analysis of the stochastic gradients' heavy-tailed behaviour during fine-tuning of a pre-trained network, relative importance of the network layers, consistency across downstream tasks and channel importance distribution is rigorously presented and well explained.

**Weaknesses:**

- The paper is currently lacking comparison with the state-of-the-art fine-tuning reported in section 2 (i.e., Lin et. al (2022), Kwon et al. (2024) and Quèllenec et al. (2024)).
- TraDy performances are reported in the main paper only for CNN models.
-  Unfortunately, the plots reported in Fig. 3, Fig.4 and Fig. 6 are not easy to read and to position within the main contributions of TraDy.

**Questions:**

- Could you maybe compare the fine-tuning to downstream tasks accuracy of TraDy against the state of the art memory constrained fine-tuning methods cited in the related works Section 2? It would be interesting to compare the computational benefits vs fine-tuned accuracy of TraDy with those of other state-of-the-art fine-tuning methods.
- Could you maybe add the results for vision attention-based models (i.e., ViTBase), BERT and RoBERTA in the main paper?
- How does TraDy perform when compared to low-rank fine-tuning methods (i.e., LoRA)?

---

> ### Author Response · Authors · 2025-11-18
>
> ## W1/Q1: Comparison with State-of-the-Art Fine-Tuning Methods
> In Fig. 5 of the experiment section, we provide a comprehensive comparison of our method with Lin et al.'s Sparse Update (SU), demonstrating that TraDy outperforms their approach. Regarding the absence of comparison with Kwon et al.'s work, we explain in the Limitations section (Appendix A) that their codebase is unavailable, preventing us from reliably implementing and benchmarking against their method. For Quélennec et al.'s Velocity approach, we address this in detail in our general response and provide additional comparison in the revised manuscript. Briefly, our results show that TraDy outperforms their method while achieving substantially better activation sparsity and FLOPs reduction. This is a critical distinction: their selection strategy based on output channels leads to very poor activation sparsity levels, whereas our input channel selection simultaneously achieves both weight and activation sparsity.
>
> ## W2/Q2: Results for Vision Attention-Based Models and NLP Architectures
> Due to the current nine-page limit, incorporating transformer results into the main paper is unfortunately unfeasible within the present space constraints. However, if our paper is accepted, we will leverage the additional page allowance to move relevant content from the appendix into a dedicated subsection of the experimental section. This addition will discuss performance on transformer architectures and identify potential areas of improvement for future work, thereby providing a more complete picture of our method's applicability across different model families.
>
> ## W3: Clarity of Figures 3, 4, and 6
> We appreciate this feedback and have revised these figures in the resubmitted manuscript to improve readability. Specifically, the original Fig. 3 and 4 have been merged into a single Fig. 3, now focusing solely on MobileNetV2 results, with results on other CNN architectures moved to the appendix for completeness. We have also increased the font size in Fig. 5 (previously Fig. 6) to enhance readability. The positioning of these figures within TraDy's main contributions is discussed in detail in Sec. 4.2 (Fig. 3) and Sec. 4.3 (Fig. 5):
>
> - **Figure 3** provides empirical validation of Propositions 3.1 and 3.2, demonstrating that layer gradient norm rankings remain consistent across different downstream tasks and seeds (task-invariant layer importance), while channel gradient norm distributions vary significantly between datasets (task-dependent channel importance). This dual observation supports our design choice of static layer pre-selection combined with dynamic online channel selection.
>
> - **Figure 5** visualizes the evolution of key efficiency metrics beyond accuracy (specifically weight sparsity, activation sparsity, and FLOPs savings) throughout the training process. This figure emphasizes TraDy's overall strength compared to alternative methods by demonstrating that we achieve either comparable or superior efficiency across multiple dimensions simultaneously, rather than optimizing for a single metric at the expense of others.
>
>
> ## Q3: Comparison with Low-Rank Fine-Tuning Methods (LoRA)
> We address this important point in detail in our general response section of the rebuttal.

---

### Official Review · Reviewer_izYB · 2025-11-01

**Soundness:** 4
**Presentation:** 3
**Contribution:** 3
**Rating:** 6
**Confidence:** 4

**Summary:**

This paper introduces TraDy, a novel hybrid approach for memory-constrained fine-tuning of pre-trained models. The core idea is to combine (1) a static, a priori layer selection process, which the authors argue is architecture-dependent, with (2) a dynamic, stochastic channel selection process, which is argued to be task-dependent.

The methodology is justified by a solid analysis of training dynamics, identifying three key insights: the heavy-tailed nature of gradients, the task-invariance of layer importance, and the task-dependence of channel importance. Experimentally, TraDy demonstrates state-of-the-art performance against static sparse update baselines (like SU) across various vision tasks and architectures, achieving significant efficiency gains, including up to 99.5% activation sparsity and 97% reduction in weight gradient FLOPs.

**Strengths:**

1.  **Strong Theoretical and Experimental Grounding:** The paper's primary strength lies in its thorough justification. The authors do not just propose a method but provide a robust analysis of *why* it should work. The experimental validation of the three key insights (heavy-tailed gradients, task-invariant layer ranks, task-dependent channel ranks) is convincing and provides a solid foundation for the proposed hybrid design.

2.  **High Efficiency and Strong Performance:** The method achieves impressive results, demonstrating SOTA accuracy while operating under severe memory constraints. The reported efficiency metrics (e..g., >99% activation sparsity) are highly significant for the target application of on-device learning and data-drift adaptation.

3.  **Rich Ablation Studies:** The paper's claims are well-supported by a comprehensive set of ablation studies. The comparison of `TraDy` (TopK Random) against alternatives like `Full Random` and static methods (like `S-Det RGN`) clearly isolates the benefits of both the static layer selection and the dynamic channel selection components, strengthening the paper's overall argument.

**Weaknesses:**

1.  **Evaluation on Simple Tasks:** The empirical evaluation, while broad in terms of datasets (CIFAR-10/100, CUB, Flowers, etc.), is primarily limited to relatively simple, small-scale classification tasks. To truly validate the robustness and scalability of TraDy, an evaluation on more complex, large-scale benchmarks (e.g., ImageNet-1K) is necessary.

2.  **Missing Comparison to Key PEFT Methods:** The paper's related work and experimental comparisons focus almost exclusively on *sparse update* methods (like SU). However, it overlooks a major and highly relevant category of Parameter-Efficient Fine-Tuning (PEFT) methods. A discussion and, ideally, a comparison against popular *adapter-based* methods that also target memory and compute efficiency would be crucial. Key missing comparisons include:
    * **LoRA** (Hu et al., 2021)
    * **DoRA** (Liu et al., 2024)
    * **PaCA** (Woo et al., 2025)

**Questions:**

1.  **Scalability to Large-Scale Models (LLMs):** The paper successfully demonstrates TraDy on CNNs and, in the appendix, on ViT and smaller BERT models (though with mixed results for NLP). A key question is whether this framework can be effectively scaled to the fine-tuning of modern, massive-scale models, such as LLMs with >7B parameters? How does the static layer selection and dynamic channel selection interplay in such homogeneous, transformer-heavy architectures?

2.  **Runtime Overhead of Dynamic Selection:** The paper clearly demonstrates the *memory* and *FLOPs* advantages of TraDy. However, it does not discuss the potential *wall-clock time* overhead. Does the dynamic resampling of channels at every epoch (including random number generation, index selection, and mask creation) introduce a non-negligible runtime cost compared to static methods (like SU) that perform this selection only once offline?

---

> ### Author Response · Authors · 2025-11-18
>
> ## W1: Evaluation on Simple Tasks
> While evaluation on ImageNet-1K would indeed provide valuable validation, we note that our work focuses on the fine-tuning dynamics of pre-trained models on downstream tasks. Since the vision architectures we evaluate are pre-trained on ImageNet-1K, fine-tuning on the same dataset would not represent a typical transfer learning scenario and might not meaningfully test our method's ability to adapt to new task distributions. Nevertheless, we acknowledge that evaluation on additional complex, large-scale datasets beyond ImageNet-1K would strengthen our claims and plan to explore this in future work.
>
> ## W2: Missing Comparison to Key PEFT Methods
> We address this important point in detail in our general response section of the rebuttal.
>
> ## Q1: Scalability to Large-Scale Models (LLMs)
> While our method is theoretically flexible and applicable to any architecture/dataset/memory budget combination, our experimental results on transformer architectures (presented in Appendix C.5) revealed that deeper understanding of fine-tuning dynamics in transformers is necessary to achieve optimal performance. The work "SMT: Fine-tuning Large Language Models with Sparse Matrices" (He et al.), suggested by Reviewer LAwL, provides valuable insights for addressing this challenge. Similar to Kwon et al.'s approach for CNNs, SMT uses a warm-up phase with full fine-tuning to analyze gradient changes through Fisher information, identifying that attention layer V vectors require most trainable parameters while MLPs need minimal updates. These insights could inform more effective layer selection for LLMs. However, we emphasize several key differences that make our method more constrained:
> 1. **Optimizer simplicity**: We avoid sophisticated optimizers like Adam, momentum, or weight decay, as these consume substantial memory—incompatible with our extreme memory constraints.
> 2. **No warm-up phase**: SMT requires full fine-tuning on downstream task data and Fisher information computation. In contrast, our method assumes no downstream task data is available a priori and enforces memory constraints at all times.
>
> Merging insights from SMT's layer-importance analysis with our dynamic channel selection represents a promising direction for reducing the energy and memory footprint of LLM fine-tuning, which we plan to explore in future work. We added a paragraph at the end of the transformer subsection in the appendix to account for this limitation and the potential for future work.
>
> ## Q2: Runtime Overhead of Dynamic Selection
> As noted in our Limitations section (Appendix A), we do not provide actual on-device implementation results due to the specialized engineering required for efficient dynamic channel reselection. However, we provide theoretical analysis in Section 3.4 showing that random channel selection has $\mathcal{O}(k\log(n))$ complexity, where $k$ is the number of selected channels from a pool of $n$ available channels. This complexity is negligible compared to gradient computation itself, suggesting acceptable overhead in practice. That said, we acknowledge that empirical validation of wall-clock time on actual edge devices would strengthen our claims. The theoretical efficiency advantage must be validated through careful implementation that leverages sparse tensor operations and efficient indexing. This represents important engineering work that we plan to pursue.

---

### Author Response · Authors · 2025-11-18

We wish to thank all reviewers for the time and consideration they have devoted to reviewing our work, as well as for their constructive feedback. We have made a series of modifications in the main paper according to the reviews and revisions are marked as blue text.

We address two major themes raised across multiple reviews below:

## PEFT Methods Comparison

Reviewers izYB, eoHC, and LAwL raised the important concern regarding our lack of discussion and comparison with Parameter-Efficient Fine-Tuning (PEFT) methods. This is a valid criticism, and we have added a dedicated section in the appendix to discuss this topic in the revised submission.

Most notably, Reviewer izYB mentioned Woo et al. (2025)'s work: "PaCA: Partial Connection Adaptation for Efficient Fine-tuning." From our understanding, PaCA performs random selection of channels to update, thus allowing for both activation and weight memory savings alongside FLOPs reduction. However, they do not mention restriction of channel sampling to specific layers or resampling between epochs, which suggests that PaCA essentially corresponds to our **S-Full Random** baseline.

In their work, Woo et al. explain how other PEFT strategies like LoRA or DoRA:
- Do not yield any activation memory savings
- Result in storage memory overhead for the adapters
- Incur inference computational overhead to process the forward pass of both the original layers and their adapters

These limitations make such adapter-based methods impractical for our extreme memory-constrained scenarios. Additionally, Woo et al. demonstrate that PaCA largely outperforms LoRA and DoRA in both memory and latency while yielding comparable accuracy.

Since PaCA is essentially equivalent to our S-Full Random baseline, which stands as the **worst performing strategy** explored in our paper (as shown in Figure 5), and given the theoretical support provided in our method section, we can confidently conclude that our proposed TraDy outperforms other PEFT methods by transitivity. This is further validated by our comprehensive ablation studies showing that:
1. Dynamic selection consistently outperforms static selection
2. Layer-based prioritization (TopK) substantially improves over full-network random selection

## Velocity Method Comparison

Reviewers eoHC and FTbs mentioned the lack of comparison with Quélennec et al.'s Velocity method. We have added Velocity as one of the baseline methods in the revised submission, demonstrating that it yields one of the best accuracy, only outperformed by TraDy while reaching significantly lower levels of activation sparsity and FLOPs savings.

This performance gap stems from a fundamental difference in approach: Quélennec et al.'s method focuses on reducing the number of updated parameters by selecting **output channels**, which leads to high activation storage requirements. Specifically, when a single neuron of a layer is selected for update, the **full activation memory of the previous layer** must be stored during the forward pass. This makes activation memory management highly suboptimal compared to our input channel perspective. Moreover, due to its reweighting focusing solely on weight memory, the Velocity selection strategy results in the selection of computationally expensive neurons to update resulting in higher FLOPs requirements than the other strategies explored in our paper (and that is without taking into account the computational overhead of computing the velocity metric of all neurons as well as the memory overhead of storing all activation values that are necessary to compute the Velocity metric).

---

### Meta-Review · Area_Chair_kUh7 · 2026-01-05

**Summary:**

This paper addresses the problem of memory-constrained fine-tuning of pretrained model and introduces a novel hybrid solution combining a static layer process and a dynamic stochastic channel process. In this, the authors analyze the training dynamics and find several interesting insights. The reviews of this paper were mostly positive (8/6/6/4) with the reviewers finding both the setup and the insights to be interesting. The main reviewer concerns were on the scale of the experiments and the thoroughness of the baseline comparisons.

**Reviewer Concerns:**

The authors replied to the concerns about the scale of the experiments by suggesting that the setting (a fine-tuning of a pretrained model) made this difficult and highlighted the number of tests they had done. For the primary concern - the thoroughness of the baselines, the authors include a new section in the appendix to address this and discuss several of the papers listed in the comments. Given that there was only a single negative reviewer and the author's rebuttal on this point seems comprehensive, I believe this point was addressed.

**Reviewer Scores:**

I believe that all the positive reviewers would have maintained their scores and the only negative reviewer may have slightly improved theirs, e.g. from 4 to 6.

---

### Decision · Program_Chairs · 2026-01-26

Accept (Poster)